# Data-driven surrogate modeling of high-resolution sea-ice thickness in the Arctic

Charlotte Durand[1], Tobias Sebastian Finn[1], Alban Farchi[1], Marc Bocquet[1], Guillaume Boutin[2], and Einar Ólason[2]

[1]CEREA, École des Ponts and EDF R&D, Île-de-France, France
[2]Nansen Environmental and Remote Sensing Center, 5007 Bergen, Norway

**Correspondence:** Charlotte Durand (charlotte.durand@enpc.fr)

**Abstract.** A novel generation of sea-ice models with Elasto-Brittle rheologies, such as neXtSIM, can represent sea-ice processes with an unprecedented accuracy at the mesoscale, for resolutions of around $10\,\mathrm{km}$. As these models are computationally expensive, we introduce supervised deep learning techniques for surrogate modeling of the sea-ice thickness from neXtSIM simulations. We adapt a convolutional UNet architecture to an Arctic-wide setup by taking the land-sea mask with partial

convolutions into account. Trained to emulate the sea-ice thickness on a lead time of 12 hours, the neural network can be iteratively applied to predictions up to a year. The improvements of the surrogate model over a persistence forecast persist from 12 hours to roughly a year, with improvements of up to $50\,\%$ in the forecast error. Moreover, the predictability gain for the sea-ice thickness measured against daily climatology extends to over 6 months. By using atmospheric forcings as additional input, the surrogate model can represent advective and thermodynamical processes, which influence the sea-ice thickness and

the growth and melting therein. While iterating, the surrogate model experiences diffusive processes, which result into a loss of fine-scale structures. However, this smoothing increases the coherence of large-scale features and hereby the stability of the model. Therefore, based on these results, we see a huge potential for surrogate modeling of state-of-art sea-ice models with neural networks.

## 1  Introduction

Sea-ice models are used to simulate and predict the changes in sea-ice cover and its effects on the Arctic and global climate. These models are based on a combination of observational data and theoretical understanding of the physical processes that govern sea-ice dynamics. They are essential conceptual and numerical tools to understand the past, current and future state of the Arctic sea-ice, and to identify the key processes that drive its changes.

Here, we present a novel way to make use of data coming from theoretical understanding of the physical processes: based

on neural networks, we build a surrogate model for the sea-ice thickness as simulated by the Arctic-wide neXtSIM model (Rampal et al., 2016; Ólason et al., 2022).

Several sea-ice models, like CICE (Hunke et al., 2017) and SI3 (Sievers et al., 2022), are concurrently developed for operational purposes: short-term predictions for maritime route and weather forecast as well as long-term simulations for climate

projections. The recent development of models based on brittle rheologies (Girard et al., 2011; Rampal et al., 2016; Dansereau et al., 2016), like neXtSIM, can represent the observed effects of small-scale processes onto the resolved mesoscale with $\sim 10\,\mathrm{km}$ horizontal resolution (Bouchat et al., 2022). Small-scale sea-ice dynamics also impact the global sea-ice mass balance (Boutin et al., 2023). Divergent features in the ice, like leads, are associated with localized intense ocean heat loss that enhances sea-ice production in winter (Kwok, 2006; von Albedyll et al., 2022), accounting for about $30\%$ of the total ice production in the Arctic Ocean. The assumption that models correctly representing the effects of such small-scale processes could have also an advantage in representing the thermodynamics of sea ice is an ongoing topic of research.

Geophysical models are computationally expensive, especially for operational forecasts. However, geophysical models can be partially or completely emulated using data-driven surrogate models. Such surrogate models can speed up the forecasting process, once their costly training phase is finished. Notably, the development of more powerful graphics processing units (GPU) in the past few years favors the use of neural networks for surrogate modeling.

Over the past years, emulating or replacing geophysical models by neural networks has become a promising topic of research, with recent overviews by Bocquet (2023); Cheng et al. (2023). Emulating ERA5 data (Hersbach et al., 2020), the European Center for Medium-Range Weather Forecasts (ECMWF) reanalysis product, recent examples of global-scale surrogate models adopt developments from computer vision by using graph neural networks (Keisler, 2022; Lam et al., 2022) and vision transformers (Bi et al., 2022; Nguyen et al., 2023).

By employing convolutional neural network architectures, Liu et al. (2020) and Andersson et al. (2021) have successfully shown that probabilistic sea-ice concentration and sea-ice extent can be predicted in a probabilistic way. Furthermore, Horvat and Roach (2022) and Finn et al. (2023b) have recently presented neural network approaches to emulate wave-ice interactions and high-resolution sea-ice dynamics. Convolutional LSTM-based neural networks have previously been investigated by Liu et al. (2021a, b); Kim et al. (2020) for sea-ice concentration forecasts.

Encouraged by such examples, we introduce a neural network to emulate the sea-ice thickness from Arctic-wide neXtSIM simulations. Using a convolutional U-Net architecture, we train the network to predict the thickness for a lead-time of 12 hours based on initial thickness conditions and atmospheric forcings. This surrogate model can be then sequentially applied to obtain sea-ice thickness predictions for seasonal time-scales.

We concentrate the surrogate model on the sea-ice thickness, as it is an important quantity for the forecast of sea ice and, yet, difficult to predict, especially on short time-scales (Zampieri et al., 2018; Xiu et al., 2022). Nonetheless, the thickness contains useful information for seasonal forecast (Balan-Sarojini et al., 2021) with direct links to other important quantities, like the sea-ice concentration and sea-ice extent.

Our surrogate model is trained to minimize the $\mathcal{L}_2$ error. This type of error metric tends to smooth out features of the fields that lead to double penalty errors, like leads in sea ice. This diffusion process has been previously observed for deterministic neural networks (Ravuri et al., 2021), when optimized on $\mathcal{L}_2$ error, but also within many forecasting and data assimilation problems in geosciences (e.g., Amodei and Stein, 2009; Farchi et al., 2016; Vanderbecken et al., 2023). To quantify the diffusion, we propose in this paper an analysis based on the Power Spectral Density (PSD).

Section 2 introduces the dataset from which we train the data-driven model and its structure. Section 3 presents the neural network framework, the choices we made about its architecture, and its optimization. Section 4 introduces the metrics for evaluating the results of the surrogate model. Section 5 delivers and discusses the results of the neural networks training, forecast skill abilities and advection capabilities of the surrogate model, as well as an analysis of the diffusion phenomenon introduced above. The discussions and conclusions are given in Section 6 and Section 7. Appendices provide technical information and further illustrations of the results.

## 2   Description of the dataset

Our goal is to train a neural network to emulate the sea-ice thickness (SIT) for a lead time of $12\,\mathrm{h}$. As training dataset, we extract the SIT from neXtSIM simulations and atmospheric forcings from the ERA5 reanalysis, which we introduce in the following.

### 2.1   neXtSIM model and the sea-ice thickness

neXtSIM is a dynamic and thermodynamic sea-ice model (Rampal et al., 2016). It currently uses Brittle Bingham-Maxwell rheology (Boutin et al., 2023; Ólason et al., 2022) to emulate the mechanical behavior of sea ice. neXtSIM can represent the observed fine-scale dynamics of sea ice, including its scaling and multifractal properties in space and in time (Rampal et al., 2019; Bouchat et al., 2022). The model is discretized on a Lagrangian triangular mesh. The model output is projected on a static quadratic grid, on which our surrogate model is based. The sea-ice model is coupled with the ocean part of NEMO, OPA, (version 3.6, Madec et al., 1998; Rousset et al., 2015). The model configuration is further detailed in Appendix A.

In this study, we extract only the sea-ice thickness, the variable predicted by the neural network. We rely on simulations from 2006 to 2018. As simulation model area, the simulations use the regional CREG025 configuration (Talandier and Lique, 2021), a regional extraction of the global ORCA025 configuration developed by the Drakkar consortium (Bernard et al., 2006). This area encompasses the Arctic and parts of the North Atlantic down to $27°N$ latitude with a nominal horizontal resolution of $0.25°$ ($\simeq 12\,\mathrm{km}$ in the Arctic basin). The outputs projected onto the static grid can be seen as two-dimensional images with $603 \times 528$ grid-cells. Without loss of information, we can crop the data to $512 \times 512$ grid-cells: lower latitudes are removed, as well as zones in Eastern Europe and America, where no sea ice appears. An example of the simulated sea-ice thickness snapshot is presented in Fig. 1.

### 2.2   Forcing fields

Several atmospheric forcings are added as input fields to the neural network, as the dynamics of the sea-ice thickness are especially driven by the atmosphere (Guemas et al., 2014): the subseasonal to interannual variability in the Arctic surface circulation is predominantly influenced by patterns in the atmospheric wind (Serreze et al., 1992). The atmospheric winds play a crucial role in shaping and driving the circulation patterns of the Arctic Ocean, which in turn affects the movement and

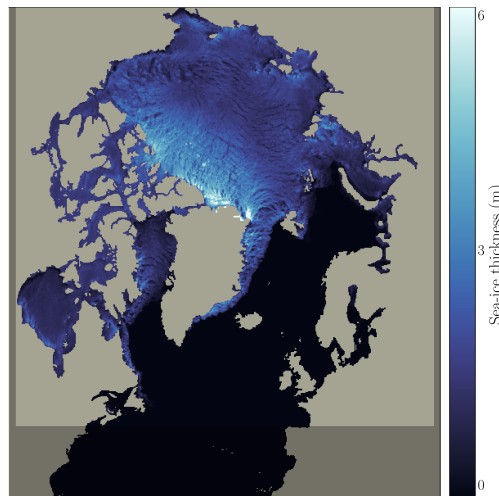

**Figure 1.** SIT simulated by neXtSIM at 15:00 UTC on 03/03/2009. The shaded area represent the cropped grid-cells that are further removed in order to keep a $512 \times 512$ grid cell SIT field without loss of information

distribution of sea ice. Additionally, fluctuations in the atmospheric surface temperature have a significant impact on the Arctic sea ice variability (Olonscheck et al., 2019). Changes in atmospheric temperature directly affect the growth, melt, and overall state of sea ice in the Arctic region. Warmer atmospheric temperatures accelerate sea ice melting, leading to reductions in ice extent and thickness, while colder temperatures can promote ice growth and expansion.

Based on these considerations, we supplemented to the sea-ice thickness the $2\,\mathrm{meters}$ temperature (T2M), and the atmospheric $u$- and $v$-velocities at a height of $10\,\mathrm{meters}$ (U10 and V10). Those forcings come from the ERA5 reanalysis dataset.

ERA5 forcings are interpolated on neXtSIM Lagrangian grid using a nearest neighbors scheme. Furthermore, to guide the temporal development of the sea ice, forcings at time $t + 6\mathrm{h}$ and $t + 12\mathrm{h}$ are added as predictors to the neural network, as commonly done in sea-ice forecasting (Grigoryev et al., 2022). We chose to incorporate future forcings based on the understanding that, in sea-ice modeling, the evolution of sea ice is strongly influenced by the atmospheric forcings. In the simulations on which our dataset is based on (Boutin et al., 2023), neXtSIM is uncoupled from an atmospheric model and uses ERA5 forcings. In such uncoupled settings, the atmospheric forcing can be given by forecasts, and, thus, known for the future. Consequently, using future forcings during training is nonrestrictive in terms of its potential operational capability.

Let us note that for neXtSIM simulations, the atmospheric forcings consist of the two $10\,\mathrm{meters}$ wind velocity components, the $2\,\mathrm{meters}$ temperature, the mixing ratio, the mean sea level pressure, the total precipitation, and the snow fraction. We decided to limit ourselves to the first three for our surrogate model. Plueddemann et al. (1998) and Kwok et al. (2013), for example, have shown that the sea-ice drift is strongly linked to the wind velocity. Hence, there exists a strong correlation between the atmosphere winds and the sea-ice motion, up to $0.8$ in Central Arctic (Thorndike and Colony, 1982; Serreze et al., 1989; Zhang et al., 2000). Those forcings can be a good proxy for the advection of sea ice, which is required to correctly emulate

the dynamics of the sea-ice thickness. Note however that T2M forcings from ERA5 are known to have an important bias during the freezing period (Yu et al., 2021; Wang et al., 2019; Køltzow et al., 2022; Nielsen-Englyst et al., 2021). Nonetheless, to stay as close to the configuration of neXtSIM simulations, we maintain the ERA5 reanalysis as forcings. We thus assume a perfect knowledge of the forcings, albeit operational sea-ice forecasts use atmospheric forecasts as forcings.

## 3 Learning the dynamics of sea-ice thickness with neural networks

In this section, we provide a description of the neural network structure, its input and output, the training process, and the various neural networks that were trained. During training, the neural network is trained in a supervised setting. The input to the network consists of the concatenated sea-ice thickness and atmospheric fields, whereas the predicted target is the increment in sea-ice thickness over the subsequent $12\,\mathrm{h}$ period. One challenge in training the neural network is dealing with unavailable data points caused by land grid-cells. To address this challenge, a technique called partial convolution is employed.

### 3.1 Preparation of the dataset for supervised learning

Let us represent the sea-ice thickness at time $t$ by $\mathbf{x}_t \in \mathrm{R}^{512 \times 512}$. The land-masked grid-cells are systematically assigned a value of zero thickness. For small signal levels, the noise induced by the imperfections of the neural network can overshadow the signal contained in the data. Consequently, to increase the signal in the target and decrease the auto-correlation, we chose a lead time of $12\,\mathrm{h}$, even though the data is available at a $6\,\mathrm{h}$ frequency.

The neural network is trained to predict the increment in SIT instead of the absolute SIT. The increments of the SIT $\mathbf{y}_{t+\Delta t}$ for $\Delta t = 12\,\mathrm{h}$ are given by the difference to a persistence forecast,

$$\mathbf{y}_{t+\Delta t} \triangleq \mathbf{x}_{t+\Delta t} - \mathbf{x}_t. \tag{1}$$

Based on the current SIT $\mathbf{x}_t$ and given forcings $\mathbf{F}$, our objective is to construct a neural network $f_{\boldsymbol{\theta}}(\mathbf{x}_t, \mathbf{F})$ with its parameters $\boldsymbol{\theta}$, which predicts the SIT increment $\mathbf{y}_{t+\Delta t}^{\mathrm{f}}$,

$$\mathbf{y}_{t+\Delta t}^{\mathrm{f}} = f_{\boldsymbol{\theta}}(\mathbf{x}_t, \mathbf{F}). \tag{2}$$

The neural network is hereby trained to approximate the real increment estimated from neXtSIM simulations, Eq. (1) such that $\mathbf{y}_{t+\Delta t} \approx \mathbf{y}_{t+\Delta t}^{\mathrm{f}}$ approximately holds.

A table detailing the inputs and target for the neural network is shown in Tab. 1. In order to represent the temporal development of the sea-ice thickness with the neural network, we also add to the inputs the fields at time $t - \Delta t$, both SIT and the atmospheric forcings. When the neural networks are trained on those fields at time $t - \Delta t$ and $t$, there are called later 'with 2 inputs'. Otherwise, the neural networks are trained 'with 1 input' which correspond to the last three columns of the inputs described in the table.

Data from 2009 to 2016 is used for training, giving 11584 training samples. 2017 is used for the validation of the learned neural network and all the preliminary tests of the surrogate model. 2018 is used as year for testing: the results were evaluated

**Table 1.** Inputs and targets for the neural networks. The table shows the predictors, including sea-ice thickness (SIT) at different time steps, and atmospheric variables: $2\,\mathrm{meters}$ temperature (T2M), $10\,\mathrm{meters}$ wind components (U10 and V10). The target is the increment of the SIT $12\,\mathrm{h}$ later ($\Delta t = 12\,\mathrm{h}$). The evaluated neural networks use either the last 3 columns as input, learning with a single timestep for SIT ($x_t$), later called neural networks with 1 input, or all columns, learning with both $x_{t-\Delta t}$ and $x_t$, later called neural networks with 2 inputs. Note, as the SIT is the predicted quantity, there are no SIT values in the inputs for time steps larger than $t$.

| Inputs | | | | Target |
|---|---|---|---|---|
| $\mathrm{SIT}(t-\Delta t)$ | $\mathrm{SIT}(t)$ | - | - | |
| $\mathrm{T2M}(t-\Delta t)$ | $\mathrm{T2M}(t)$ | $\mathrm{T2M}(t+\Delta t/2)$ | $\mathrm{T2M}(t+\Delta t)$ | $\Delta\,\mathrm{SIT}(t+\Delta t)$ |
| $\mathrm{U10}(t-\Delta t)$ | $\mathrm{U10}(t)$ | $\mathrm{U10}(t+\Delta t/2)$ | $\mathrm{U10}(t+\Delta t)$ | |
| $\mathrm{V10}(t-\Delta t)$ | $\mathrm{V10}(t)$ | $\mathrm{V10}(t+\Delta t/2)$ | $\mathrm{V10}(t+\Delta t)$ | |

NN 2 inputs

NN 1 input

once on this year at the end of the study, after the hyperparameters were chosen for the neural network. For longer forecasts, to evaluate seasonal forecasts, another test dataset was built from the years 2006, 2007 and 2008.

The input and target data are normalized by a global per-variable mean and standard deviation. These statistics are estimated over the entire training dataset and applied to all datasets.

### 3.2 Neural network architecture

Convolutional neural networks (CNN) are largely used in computer vision and have shown to be scalable to high-dimensional datasets (e.g., Pinckaers et al., 2022). These networks are based on convolutional layers, designed to recognize translation-invariant patterns. In the case of sea-ice thickness, the neural networks need to detect, e.g., leads, as well as the marginal ice zone, irrespective of their actual locations.

The UNet (Ronneberger et al., 2015) is an encoder-decoder convolutional neural network architecture with skip-connections. In the encoding part, convolutional layers and max-pooling layers are stacked in order to extract spatially more and more compressed features. As convolutional layers are localized by definition, the spatial compression helps the network to extract more globalized features. The number of successive resolution reduction defines the depth of the UNet. At the lowest resolution, the bottleneck, several convolutional layers are stacked with 256 features (channels). In the decoding part, the features are up-sampled through a nearest neighbor interpolation and convolutional layers. Skip connections couple the encoding and decoding part at the same resolution level to facilitate training and to retain fine-granular information in the network. This neural network architecture is designed to extract multiscale features, which is known to be notably present in sea-ice dynamics (Rampal et al., 2019). The UNet used here is described in detail in Appendix C and schematically outlined in Fig. 2.

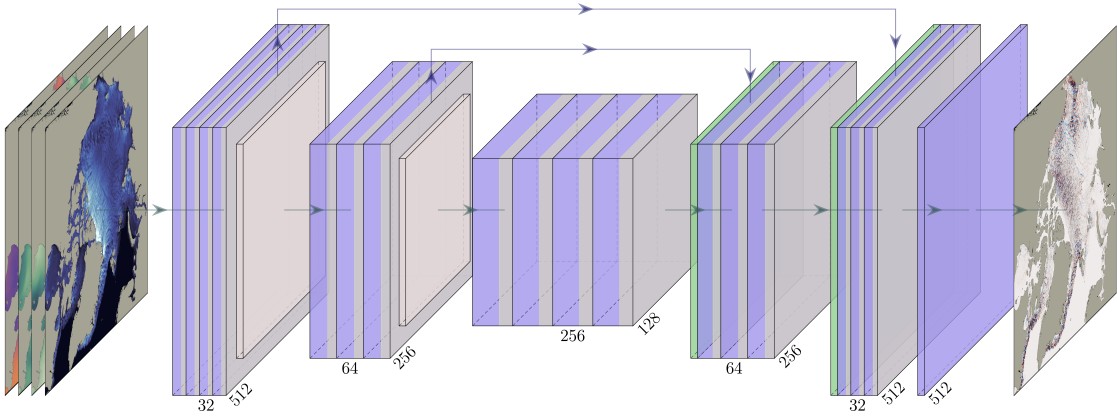

**Figure 2.** Architecture of the UNet-based neural networks. The UNet consists of three levels of depth with image sizes of 512, 256, and 128, in $x$- and $y$-direction. The input of the UNet includes either 10 or 14 channels, depending on whether only the current time step ($x_t$) or both the current and previous time steps ($x_t$ and $x_{t-\Delta t}$) are used, alongside their associated atmospheric forcings. The input channels comprise sea-ice thickness, air velocities, and temperature. The number of channels for each convolution is indicated below it, with the first block having 32 channels. The upward arrows represent skip connections, allowing the neural network to retain information from earlier stages and incorporate it into subsequent stages, bypassing the bottleneck.

The last layer of the neural network is a linear function without any activation, as we cast learning of the SIT increment as a regression problem. For all other layers, the *mish* activation function (Misra, 2019) is used. As opposed to the more-often used rectified linear unit, mish is a continuously differentiable function and has been previously proven to be effective in computer vision tasks (Bochkovskiy et al., 2020; Zhang et al., 2019), demonstrating improvements in training CNNs, particularly in addressing issues such as gradient explosion and gradient dispersion.

### 3.3 Partial convolution

As we can see in Fig. 1, the information on sea-ice thickness is only defined for ice and ocean grid-cells. Land grid-cells are masked. When performing two-dimensional convolutions on land cells, the presence of masked values has a detrimental effect on the local averages computed during the convolution operation. The convolution kernels then also includes the land cells with an assigned value of 0. One solution is to use partial convolutions (Liu et al., 2018) in every convolutional layer of the neural network. The key idea of partial convolutions is to separate the missing points from informative ones during convolutions, such that the results of convolutions only depend on ocean and ice grid cells; land grid cells are simply omitted in the convolutional kernel. Let us see how it works in a simple example for a single convolution window.

Let us define $\mathbf{W} \in \mathbb{R}^{k_s \times k_s}$ and $b \in \mathbb{R}$, as the weights and bias of a convolution filter. $k_s$ is the kernel-size of each convolution, always set to 3, except for the last layer of the neural network where it is set to 1. $\mathbf{X} \in \mathbb{R}^{k_s \times k_s}$ represents the pixel values (or

feature activation values) being convoluted and $\mathbf{M} \in \mathbb{R}^{k_s \times k_s}$ is the corresponding binary mask which indicates the validity of each pixel/feature value: 0 for missing (land) pixels and 1 for valid (ocean and ice) pixels. The output of the proposed partial convolution $x' \in \mathbb{R}$, computed in a convolution window, is then

$$
\quad x' = \begin{cases} \mathbf{W}^{\mathsf{T}}(\mathbf{X} \odot \mathbf{M})\frac{\mathrm{sum}(\mathbf{1})}{\mathrm{sum}(\mathbf{M})} + b & \text{if } \sum_{i,j} \mathbf{M}_{i,j} > 0 \\ 0 & \text{otherwise,} \end{cases} \tag{3}
$$

where $\odot$ is an element-wise multiplication and $\mathbf{1}$ is a matrix of ones that has the same shape as $\mathbf{M}$. In comparison, a normal convolution would be defined as

$$
x' = \mathbf{W}^{\mathsf{T}}\mathbf{X} + b, \tag{4}
$$

independent of the validity of the grid cells.

From Eq. (3), we can see that the results of the partial convolution only depend on the valid input values (as $\mathbf{X} \odot \mathbf{M}$). The scaling factor $\mathrm{sum}(\mathbf{1})/\mathrm{sum}(\mathbf{M})$ adjusts the results as the number of valid input values for each convolution varies. It has been used previously in order to recover missing regions from observational datasets (Kadow et al., 2020). In this study, the goal is not to recover data, but to avoid artifacts near land caused by the underestimation in normal convolutions. The algorithm for partial convolution is further described in Appendix C1.

**3.4 Global constraint on loss training**

Emulating physical systems with neural networks can lead to a non-physical response (Beucler et al., 2021). In order to reduce a systematic bias of the surrogate model and to ensure that the neural network can correctly predict the global amount of sea ice, we add an additional penalization term to the loss. The non-penalized loss is defined by a pixel-wise mean-squared error (MSE), with $\mathbf{x}$ and $\mathbf{y}$ two vectors of dimension $(N_x, N_y)$

$$
\quad \mathcal{L}_{\mathrm{local}}(\mathbf{x}, \mathbf{y}) = \mathrm{MSE}(\mathbf{x}, \mathbf{y}) = \frac{1}{N_x \cdot N_y} \sum_{i,j}^{N_x, N_y} (x_{i,j} - y_{i,j})^2. \tag{5}
$$

The penalization term is defined by the squared difference between the global mean of $\mathbf{x}$ and $\mathbf{y}$:

$$
\mathcal{L}_{\mathrm{global}}(\mathbf{x}, \mathbf{y}) = (\bar{\mathbf{x}} - \bar{\mathbf{y}})^2 = \left( \frac{1}{N_x \cdot N_y} \sum_{i,j}^{N_x, N_y} x_{i,j} - y_{i,j} \right)^2. \tag{6}
$$

This term weighted against the local loss with the help of a scalar $\lambda$

$$
\mathcal{L}(\mathbf{x}, \mathbf{y}) = \mathcal{L}_{\mathrm{local}}(\mathbf{x}, \mathbf{y}) + \lambda \mathcal{L}_{\mathrm{global}}(\mathbf{x}, \mathbf{y}). \tag{7}
$$

Let us note that $\mathcal{L}_{\mathrm{local}}$ refers to the local dynamics of the sea-ice thickness, and that $\mathcal{L}_{\mathrm{global}}$ refers to the global dynamics of the sea-ice thickness.

   $\lambda$ is manually tuned to 100. Details on how this value was set are provided in Sec. C3.2. The local loss is approximately 4 orders of magnitude larger than the global loss. By setting $\lambda = 100$, the global loss represents $1\%$ of the local loss. In the following parts, we show results for $\lambda = 0$ and $\lambda = 100$. The former case will be called *unconstrained* and the latter *constrained*.

## 3.5 Neural network training

The neural networks are trained on a single NVIDIA A100 GPU with a batch size of eight samples. As optimizer, AdamW (Loshchilov and Hutter, 2017) is used with a learning rate of $\gamma = 5 \times 10^{-5}$ and a weight decay, scheduled with a 3 steps piecewise constant decay, starting at $w = 1 \times 10^{-6}$. If the loss in the independent validation dataset plateaus for 20 epochs, the training is stopped early.

We trained four different neural networks, as described in Table 2. By setting $\lambda$ to either 0 or 100, we switch the additional loss function constrain on or off, checking its influence on the performance. Additionally, we test if additional temporal guidance by giving an additional time step as input helps the neural network to predict the increment in the sea-ice thickness.

**Table 2.** Comparison of the trained UNet-based neural networks in the study. Four neural network configurations are evaluated, varying in the number of inputs and the presence of a constraining term in the loss function. The inputs include either $x_t$ alone or both $x_{t-\Delta t}$ and $x_t$, representing sea-ice thickness and atmospheric variables. The addition of a constraining term in the loss function regulates the neural network based on the global sea-ice thickness.

| Neural network | Constraint ($\lambda$) | Inputs |
|---|---|---|
| NN 1 input - unconstrained | 0 | $x_t$ (10 channels) |
| NN 1 input - constrained | 100 | $x_t$ (10 channels) |
| NN 2 input - unconstrained | 0 | $x_{t-\Delta t}, x_t$ (14 channels) |
| NN 2 input -constrained | 100 | $x_{t-\Delta t}, x_t$ (14 channels) |

## 4 Surrogate modeling and evaluation methods

### 4.1 Surrogate modeling

To emulate the physical model $\mathcal{M}^{\mathrm{p}}$, we built a surrogate model $\mathcal{M}^{\mathrm{s}}$ by applying the neural network $f_{\boldsymbol{\theta}}(\cdot,\cdot)$ that predicts the sea-ice thickness increment. Initializing the model with given initial conditions, $\mathbf{x}_{t_0}$, and given forcings $\mathbf{F}$, the surrogate model propagates the sea-ice thickness forward in-time, predicting the sea-ice thickness $\Delta t = 12\,\mathrm{h}$ later,

$$\mathbf{x}^{\mathrm{f}}_{t_0+\Delta t} = \mathbf{x}_{t_0} + f_{\boldsymbol{\theta}}(\mathbf{x}_{t_0}, \mathbf{F}) = \mathcal{M}^{\mathrm{s}}(\mathbf{x}_{t_0}). \tag{8}$$

Note, for the ease of notation, we omit the forcings into the surrogate model notation $\mathcal{M}^{\mathrm{s}}$. Using the forecasted state $\mathbf{x}^{\mathrm{f}}_{t_0+\Delta t}$ as next initial conditions, we can cycle the surrogate model and predict the sea-ice thickness for longer lead-times than $\Delta t$,

$$\mathbf{x}^{\mathrm{f}}_{t_0+N\Delta t} = \underbrace{\mathcal{M}^{\mathrm{s}} \circ \mathcal{M}^{\mathrm{s}} \circ \cdots \circ \mathcal{M}^{\mathrm{s}}(\mathbf{x}_{t_0})}_{N \text{ times}} \tag{9}$$

$$= \mathbf{x}_{t_0} + f_{\boldsymbol{\theta}}(\mathbf{x}_{t_0} + f_{\boldsymbol{\theta}}(\ldots, \mathbf{F}), \mathbf{F}). \tag{10}$$

The forecast at longer lead times is consequently the initial conditions plus a recursive increment term.

Our baseline for the model comparison is constantly predicting the initial conditions without any increment, a so-called persistence forecast, i.e. the sea-ice thickness is unchanged over time. It is a commonly used baseline in sea-ice forecasting, as the auto-correlation of the sea-ice thickness in time is high up to a 1 month-lead time (Lemke et al., 1980; Blanchard-Wrigglesworth et al., 2011). We also compare the surrogate model to the daily climatology, computed on a day-of-year basis over the complete training dataset.

## 4.2 Evaluation metrics for the surrogate

The goal of the surrogate model is to predict as accurately as possible sea-ice thickness over lead times longer than $12\,\mathrm{h}$, i.e. after several iterations of the surrogate model. We define the forecast skill of the surrogate at the $k$-th iteration by computing the root-mean-squared error (RMSE) between the predicted SIT and the actual SIT as simulated by neXtSIM,

$$
\mathrm{RMSE}(k) = \frac{1}{N_\mathrm{s}} \sum_{n=1}^{N_\mathrm{s}} \sqrt{\frac{1}{N_\mathrm{valid}} \sum_{i}^{N_\mathrm{valid}} \left(\mathbf{x}^\mathrm{f}_{t_n+k\Delta t,i} - \mathbf{x}_{t_n+k\Delta t,i}\right)^2}.
\tag{11}
$$

The RMSE between the prediction $\mathbf{x}^\mathrm{f}_{n+k\Delta t}$ and the simulation $\mathbf{x}_{n+k\Delta t}$ is computed over all valid pixels $i$ of the field of size $N_\mathrm{valid}$, i.e. pixels which are not land grid-cells, for each sample $n$ of the test set containing $N_\mathrm{s}$ trajectories, initialized at time $t_n$.

The global RMSE is calculated by averaging the RMSE values obtained when the whole sea-ice thickness fields are treated as single data point. It represents the discrepancy in averaged sea-ice thickness between the prediction and the simulation. By considering the global RMSE, we can assess the performance of the surrogate model in accurately reproducing the average sea-ice thickness compared to the reference model.

In order to quantify systematic errors of the surrogate model, we compute its mean error (bias). This metric tells about the ability of the neural network to correctly estimate the total amount of sea-ice in the full domain,

$$
\mathrm{bias}(k) = \frac{1}{N_\mathrm{s}} \sum_{n=1}^{N_\mathrm{s}} \frac{1}{N_\mathrm{valid}} \sum_{i}^{N_\mathrm{valid}} \left(\mathbf{x}^\mathrm{f}_{t_n+k\Delta t,i} - \mathbf{x}_{t_n+k\Delta t,i}\right).
\tag{12}
$$

The sea-ice extent (SIE) can be derived from the sea-ice thickness. We define a threshold $\sigma_\mathrm{acc} = 0.1\,\mathrm{m}$ for the SIT (see Appendix B for its definition) above which a grid point is considered as sea ice. By obtaining a classification mask between ice and no ice, we can easily define an accuracy metric based on the SIE. Similarly to the ice-integrated edge error, defined by Goessling et al. (2016) on sea-ice concentration, we define a metric which counts the pixels where the surrogate model disagrees with neXtSIM on the presence or not of sea-ice. Two terms are defined: the first one $N_{>\sigma_\mathrm{acc}}$ indicates the number of pixels where $\mathbf{x}^\mathrm{f}_{t_n+k\Delta t}$ overestimates the presence of sea-ice, and the second one $N_{<\sigma_\mathrm{acc}}$ where the surrogate model underestimates the presence of sea-ice compared to neXtSIM. The accuracy is averaged over all $N_\mathrm{s}$ samples:

$$
\mathrm{acc}_\mathrm{SIE}(k) = \frac{1}{N_\mathrm{s}} \sum_{n=1}^{N_\mathrm{s}} \left(1 - \frac{N_{>\sigma_\mathrm{acc}}(t_n+k\Delta t) + N_{<\sigma_\mathrm{acc}}(t_n+k\Delta t)}{N_\mathrm{sea-ice\,pixels}}\right).
\tag{13}
$$

## 4.3 Quantification of the diffusion effect

Diffusion can impact the accuracy and fidelity of the surrogate model's predictions. Excessive diffusion may lead to the loss of important details and reduce the model's ability to capture complex patterns. By quantifying diffusion, we can evaluate the model's performance and how the diffusion process evolves with increasing lead time.

To analyze the smoothing of features across multiple iterations, we want to find a metric that can describe the evolution of these features across different scales. Mathematicians have proposed several metrics for quantifying multifracality, such as box-counting algorithms and fractal dimensions (Xu et al., 1993). For two-dimensional geophysical fields, the power spectral density (PSD) has the ability to detect spatial properties over the different space scales (Lovejoy et al., 2008). This quantity allows for a quantitative assessment of the changes in the features and its multiscale distribution as a function of the forecast lead time. Let $\mathbf{x}$ be a snapshot of the sea-ice thickness at a given time, either from neXtSIM or from the surrogate model. We define the PSD of $\mathbf{x}$ by:

$$P(k_h, k_v) = \|\mathrm{dft}(\mathbf{x})(k_h, k_v)\|^2, \tag{14}$$

where $\mathrm{dft}(\mathbf{x})$ is the discrete Fourier transform of $\mathbf{x}$. The PSD $P$ is indexed by the spatial wave numbers $k_h$ and $k_v$. The energy as a function of the wave vector is in turn related to $P$ via

$$E(k_h, k_v) = (P(k_h, k_v))^2. \tag{15}$$

The power-law behavior of a field's energy spectrum can be caused by the underlying self-similarity or fractal nature of the image. Fractals are patterns or objects that display similar structures and statistical properties at various scales. In the case of an image, this means that certain statistical characteristics, such as texture or pixel intensity variations, repeat themselves across different scales. This energy spectrum can be identified with a power-law,

$$E(\mathbf{k}) \sim \mathrm{C}\|\mathbf{k}\|^{-\beta}, \tag{16}$$

where $\beta$ is called *spectral exponent* and C the normalization constant. Details on the computation of the spectral exponent are provided in Appendix D5. The power-law nature of the energy spectrum reflects the scaling properties of the field, where the statistical variations remain consistent regardless of the scales being observed. The power-law exponent $\beta$ determines the degree of self-similarity and how quickly the energy decreases as the frequency or spatial scale increases.

We define the spectral exponents' ratio $Q_\beta$ after $t$ iterations by

$$Q_\beta(t) = \frac{1}{N_s} \sum_{i=1}^{N_s} \frac{\beta_{\mathrm{surr}}^i(t)}{\beta_{\mathrm{neXtSIM}}^i(t)}, \tag{17}$$

the average over the full testing set of the ratio between the spectral exponent of predicted fields from the surrogate model and the spectral exponent of the fields as simulated with neXtSIM at the same time. If the surrogate model exhibits processes which lead to over-diffusion, then the spectral exponent of the predicted SIT should be larger than that of the actual SIT, resulting into a ratio $Q_b > 1$. Hence, the ratio corroborates the emergence of diffusion in the forecast.

## 5 Numerical results

### 5.1 Short-term forecasting

In this paragraph, we assess the performance of the surrogate model on a short-term timescale, specifically up to a one-month lead time. The metrics mentioned in Sec 4.2 are computed using the 2018 test dataset. They are reported in Table 3.

We believe that the global RMSE serves as a proxy for the consistency of the surrogate model, which we define as the averaged sea-ice thickness in the domain. Based on this idea, we anticipate that the globally-constrained neural network demonstrate improved performance for forecast lead times exceeding $12$ h.

**Table 3.** Statistical indicators to assess the performance of the surrogate models. The table shows the results for two lead time scenarios: 12 hours and 15 days. Two types of surrogate models are evaluated: those with 1 input (representing sea-ice thickness at time $x_t$) and those with 2 inputs (with SIT and atmospheric forcings at time $(x_{t-\Delta t}, x_t)$). The models are trained with and without the addition of constrains, represented by a regularization parameter (Constrains). The evaluation metrics include RMSE, global RMSE, and SIE accuracy (ACC). Climatology and persistence baselines are included for comparison. Bold numbers indicate the best performing model in a given column.

| Surrogate | Constraint ($\lambda$) | 12 hours lead time | | 15 days lead time | | |
| | | RMSE $\downarrow$ | Global RMSE $\downarrow$ | RMSE $\downarrow$ | Global RMSE $\downarrow$ | ACC $\uparrow$ |
|---|---|---|---|---|---|---|
| Climatology | - | $7.75 \times 10^{-1}$ | $2.97 \times 10^{-1}$ | 0.775 | $2.97 \times 10^{-1}$ | 0.953 |
| Persistence | - | $1.28 \times 10^{-1}$ | $3.01 \times 10^{-2}$ | 0.603 | $1.01 \times 10^{-1}$ | 0.949 |
| 1 input | 0 | $8.49 \times 10^{-2}$ | $1.35 \times 10^{-2}$ | 0.406 | $\mathbf{1.43 \times 10^{-4}}$ | 0.963 |
| 1 input | 100 | $8.59 \times 10^{-2}$ | $\mathbf{8.00 \times 10^{-4}}$ | $\mathbf{0.401}$ | $1.76 \times 10^{-3}$ | $\mathbf{0.970}$ |
| 2 input | 0 | $\mathbf{7.34 \times 10^{-2}}$ | $3.73 \times 10^{-3}$ | 0.445 | $2.91 \times 10^{-3}$ | 0.966 |
| 2 input | 100 | $7.48 \times 10^{-2}$ | $3.59 \times 10^{-3}$ | 0.416 | $1.81 \times 10^{-4}$ | 0.966 |

The introduction of a global penalization term in the constrained neural network reduces the global RMSE by one order of magnitude within 12 hours compared to the absence of penalization. However, the impact of the global loss term (as defined in Eq. 6) is relatively small on the *classical* RMSE, as defined in Eq. (11), compared to the influence of including additional time steps.

On average, for the constrained neural network with one timestep as input, we observe a $33$ % improvement after a lead time of $12$ h, and a $33$ % improvement after 15 days over the persistence on the RMSE. For the constrained neural network with two timesteps in the input, we observe a $41$ % improvement after $12$ h, and a $31$ % improvement after 15 days over persistence. It is worth noting that all surrogate models exhibit significant improvements over climatology in forecasting sea-ice dynamics for a 15-day period. On average, these improvements amount to a $46$ % enhancement compared to relying solely on climatology-based predictions for RMSE. Adding the penalization term slightly increases the RMSE of the surrogate after $12$ h but improves the RMSE after 15 days (see Fig. 3a for the comparative evolution of the RMSE for 1 input up to a lead time of 25 days). The major improvement of adding the penalization term comes from the global RMSE, which is reduced by a factor 9 after $12$ h.

Regarding the 2 inputs surrogate, the impact on the global RMSE after $12\,\mathrm{h}$ is only $4\,\%$. The global RMSE is much more volatile during training, being smaller than the local RMSE. In the case of the 2 inputs NN, the unconstrained model had a given global RMSE which happened to be similar to the model trained with the constraint, despite not being trained with the global term. The fact that the results are still better in the constrained case after 15 days is also a proof that the model improves with the global loss term.

We note that the global RMSE after 15 days is better in average for the unconstrained surrogate. Yet, as seen in Fig. 3, the strong advantage of the constrained surrogate is the important reduction of the bias standard deviation, represented in transparency in panel (b). This improvement is further supported by evaluating the averaged SIT over the entire year, as illustrated in panel (c). During periods of significant sea-ice production and melting, the surrogate model with the global constraint exhibits a closer alignment with the neXtSIM output, indicating a higher level of accuracy. These findings suggest that integrating a global constraint term during the optimization process enhances the surrogate model's ability to capture and reproduce the complex dynamics associated with sea-ice formation and melting.

As the neural network is trained for a $12\,\mathrm{h}$ lead time, the first iteration of the surrogate model corresponds to its targeted lead time. In this first iteration, we observe that stacking two timesteps in the inputs of the neural network improves the surrogate by $13\,\%$ in terms of RMSE. In preliminary tests, we observed no further gain in performance with more than two timesteps as input.

The forecast skill for up to a lead time of 25 days highlights the overall improvement of the constrained surrogate model compared to the persistence forecast, Fig. 4, as similarly observed in Tab. 3. For the constrained surrogates, the two inputs surrogate gain $14\,\%$ after $12\,\mathrm{h}$ over the one input surrogate, but the results are reversed after 15 days, with a $3\,\%$ improvement for the one input surrogate, as we can also observe in the Fig. 4. Even if those results would favor the surrogate model with two inputs, biases for either 1 or 2 inputs are close to $0$ and are thus acceptable. It is expected that the neural network gives a better RMSE with more inputs in the neural network. We hypothesize that increasing the number of inputs leads to a bigger accumulation of errors being introduced in the input data when we predict with the surrogate model, since the model was non-autoregressively trained on perfect input data. In other words, as we cycle the neural network, the predictions from previous iterations are used as inputs for subsequent iterations, so that if there are errors or inaccuracies in these predictions, they can propagate and accumulate over time, potentially leading to a degradation in the quality of the outputs. As the surrogate learns the dynamics with "perfect" conditions, we increase the error of the inputs by having two timesteps as the input of the neural network after several iterations. Even if the two timesteps neural network provides better results for the first iterations and for the global RMSE, it seems more relevant to focus on longer lead times for model selection. In the next paragraphs, we only present results for a surrogate model with one timestep in the input.

The evaluation of sea-ice extent (SIE) accuracy supports and strengthens our previous findings, providing additional evidence for the reliability of the results. In particular, we observe that the constrained neural network with one timestep as input consistently outperforms other models in predicting SIE. The higher accuracy achieved by the constrained neural network with one timestep as input suggests that this configuration effectively captures the relevant information and patterns necessary for

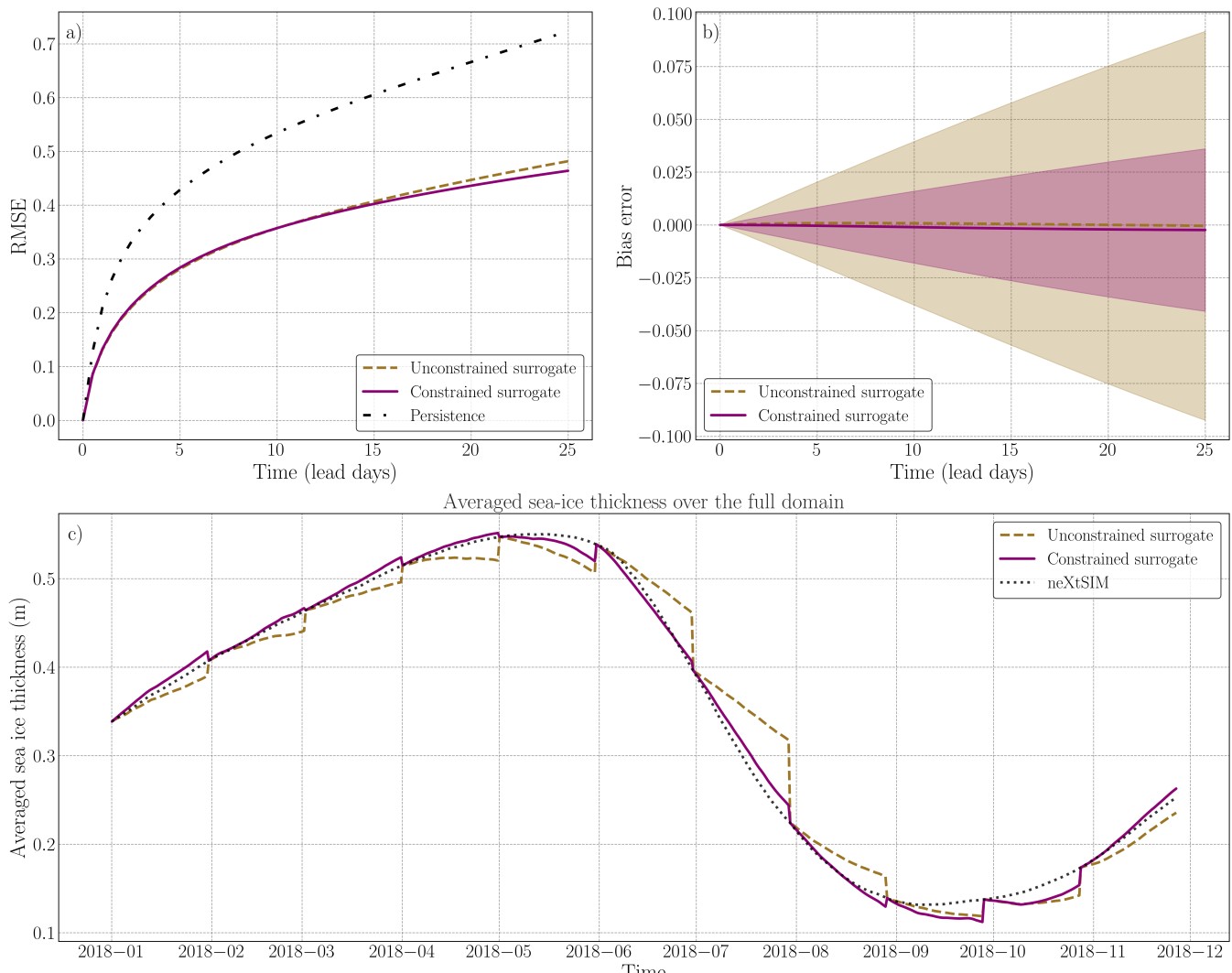

**Figure 3.** Analysis of the additional constraint during neural network training on the surrogate model over several iterations. In panel (a), the forecast skill of the surrogate model is depicted with solid lines representing the average. The unconstrained neural network is represented by the brown dashed line, while the constrained network is shown in purple. The black dotted line represents persistence. Panel (b) displays the bias error associated with the surrogate, in transparency is represented the standard deviation, to outline the variance reduction of the constrained surrogate. In panel (c), the global conservation of sea-ice is plotted. The full-year trajectory is constructed by concatenating 60 forecast iterations. Every 30 days, the forecast is initialized using neXtSIM at the corresponding time and run for 60 iterations. The surrogate models are compared to the neXtSIM output over the same period.

accurate sea-ice dynamics prediction. This indicates that the specific constraints imposed during training, along with the inclusion of a single timestep as input, contribute to a better understanding and modeling of the underlying dynamics of SIT. The

consistent performance of this model across different evaluation metrics, see Tab. 3, Fig. 3 and Fig. 4, and scenarios further validates its reliability and robustness. This surrogate configuration is able to capture the essential features and patterns of SIT

dynamics, enabling more accurate predictions compared to other configurations.

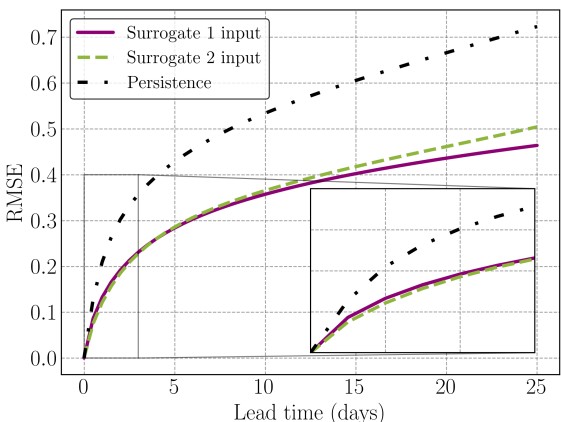

**Figure 4.** Comparison of Root Mean Square Error (RMSE) between the constrained surrogate models for either 1 (purple solid line) or 2 inputs (green dashed line), and the persistence (black dotted line) approach for sea-ice thickness (SIT) prediction over a 25-day forecast horizon. While the *1 input* surrogate yields better results in terms of RMSE for more than 5 days, the surrogate with *2 inputs* gives better results at the beginning of the forecast, as can be observed in the zoom window.

Despite its ability to predict sea-ice thickness over the full domain, the forecast skill is not homogeneous over the different regions of the Arctic. The delimitation of the region from the National Snow and Ice Data Center was interpolated on the neXtSIM grid to then compute the forecast skill on the different regions, see Fig. 5 and numerical results for 25 lead days in

Tab. 4. In Central Arctic, the surrogate forecast skill has an improvement of the RMSE of 31 % in average for a 25-day lead time over the persistence. The variability of the forecast skill is equal to 0.0590 and is 34 % lower than the variability of the persistence after the same lead-time. In Greenland Sea, the forecast skill of the surrogate is 35 % better than persistence for a 25-day lead time. The forecast skills of both the persistence and the surrogate are on average low because of the amount of sea-only pixels in this region during the full year. In every region, we systematically observe an improvement in both RMSE

and its standard deviation of the surrogate over the persistence. This means that we improve over most samples the ability of the surrogate to predict the dynamics, across all regions. Notably, the Beaufort Sea exhibits a higher RMSE compared to the other regions. This discrepancy prompts further investigation into the surrogate model's limitations in accurately predicting sea ice dynamics near land in this region.

**Table 4.** Comparison of RMSE and its standard deviation ($\sigma_{\mathrm{RMSE}}$) between the surrogate model and persistence for different regions. The table presents the mean RMSE and $\sigma_{\mathrm{RMSE}}$ values for the Central Arctic, Greenland Sea, East Siberian Sea, Kara Sea, and Beaufort Sea as defined in Fig. 5a. The RMSE are computed for a lead time of 15 days.

| | Surrogate | | Persistence | |
|---|---|---|---|---|
| Regions | RMSE | $\sigma_{\mathrm{RMSE}}$ | RMSE | $\sigma_{\mathrm{RMSE}}$ |
| Central Arctic | 0.550 | 0.0590 | 0.777 | 0.0892 |
| Greenland Sea | 0.286 | 0.0591 | 0.423 | 0.0786 |
| East Siberian Sea | 0.507 | 0.0727 | 0.754 | 0.0967 |
| Kara Sea | 0.474 | 0.1103 | 0.771 | 0.1822 |
| Beaufort Sea | 0.758 | 0.0921 | 1.140 | 0.1376 |

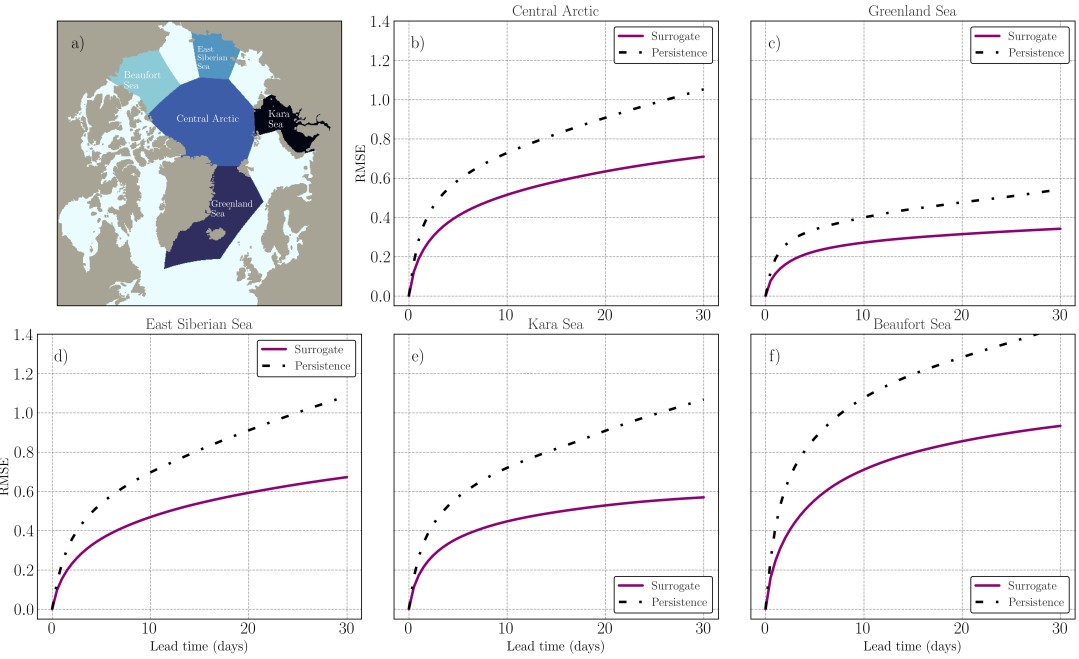

**Figure 5.** Regional forecast skills. Panel (a) illustrates the delineation of regions used for computing the regional forecast skill. Panels (b) to (f) show the averaged forecast skill over the full test year for specific regions: (b) Central Arctic, (c) Greenland Sea, (d) East Siberian Sea, (e) Kara Sea, and (f) Beaufort Sea. The purple solid line represents the surrogate model forecast skill, while the black dashed line represents persistence.

## 5.2 Advection

The surrogate model exhibits favorable advection properties, encompassing both large-scale and fine-scale dynamics. This successful advection can be attributed to the incorporation of atmospheric forcings in the model. The atmospheric forcings, which capture the influence of atmospheric conditions such as winds and temperatures, play a crucial role in driving the movement and behavior of sea ice.

The thickness field as well as the SIE are represented in Fig. 6 for neXtSIM (a) and the surrogate (b). Additional SIT fields are presented in the appendix for lead time of 5 days and lead time of 25 days in Fig. D2 and Fig. D3. The surrogate model seems to correctly advect the sea-ice sheet on the large scale. The inclusion of atmospheric forcings as inputs to the surrogate is crucial for capturing and learning the driving dynamics of sea-ice. When training the surrogate model without incorporating any atmospheric forcings, the absence of advection becomes apparent. Without the driving influence of atmospheric conditions such as winds and temperatures, the surrogate model lacks the necessary information to simulate and reproduce the advection of sea ice, it tends to exhibit behavior similar to persistence.

In order to verify this visual impression, we followed manually 4 remarkable features, (c) on the MIZ, (d) a feature in Beaufort Sea, (e) a persistent crack in central Arctic and (f) on the MIZ in the Barents Sea for 1 month prediction and compared the motion of those features between the surrogate model and the actual neXtSIM dynamics. The results depicting the sea-ice advection are illustrated in Fig. 6 in the lower panel. Notably, several features, particularly in the MIZ, demonstrate nearly identical displacements over this one-month period. These features appear to be accurately captured and reproduced by the surrogate model, reflecting its ability to simulate the advection of sea-ice. Slight deviations in trajectories are observed for features such as cracks within the sea-ice, but these differences do not indicate incoherent or erratic behavior: the trajectories keep similar paths, which could be due to the advection of the features by atmospheric forcings.

## 5.3 Diffusion quantification

The observation of a smoothing effect on fine-scale features which increases with the forecast lead time aligns with the $\mathcal{L}_2$ optimization of a deterministic neural network: although scoring well, the surrogate model can deviate from the genuine physics with a prominent smoothing of the fine scales. The goal of training is to minimize the mean square error (MSE), which entails reducing discrepancies and errors by creating an average over the features. This smoothing effect can be seen in the upper panel of Fig. 7. While the surrogate model is able to predict the global and local advection patterns of sea-ice, it tends to average out the fine-scale features over successive iterations. The observed smoothing effect highlights the trade-off between capturing large-scale dynamics and preserving fine-scale features in the surrogate model. While the model may sacrifice some fine-scale details, it still retains the essential advection patterns and provides reliable predictions on a global scale.

This smoothness can be quantified by computing the power spectral density (PSD) (Hess et al., 2023; Neuhauser et al., 2022) and the $Q_b$ ratio as defined in Sect. 4.3. The results are presented in Fig. 8. After 12 hours, the PSD of neXtSIM and the surrogate are close: the surrogate model exhibit similar multiscale properties than the physical model. We found the spectral

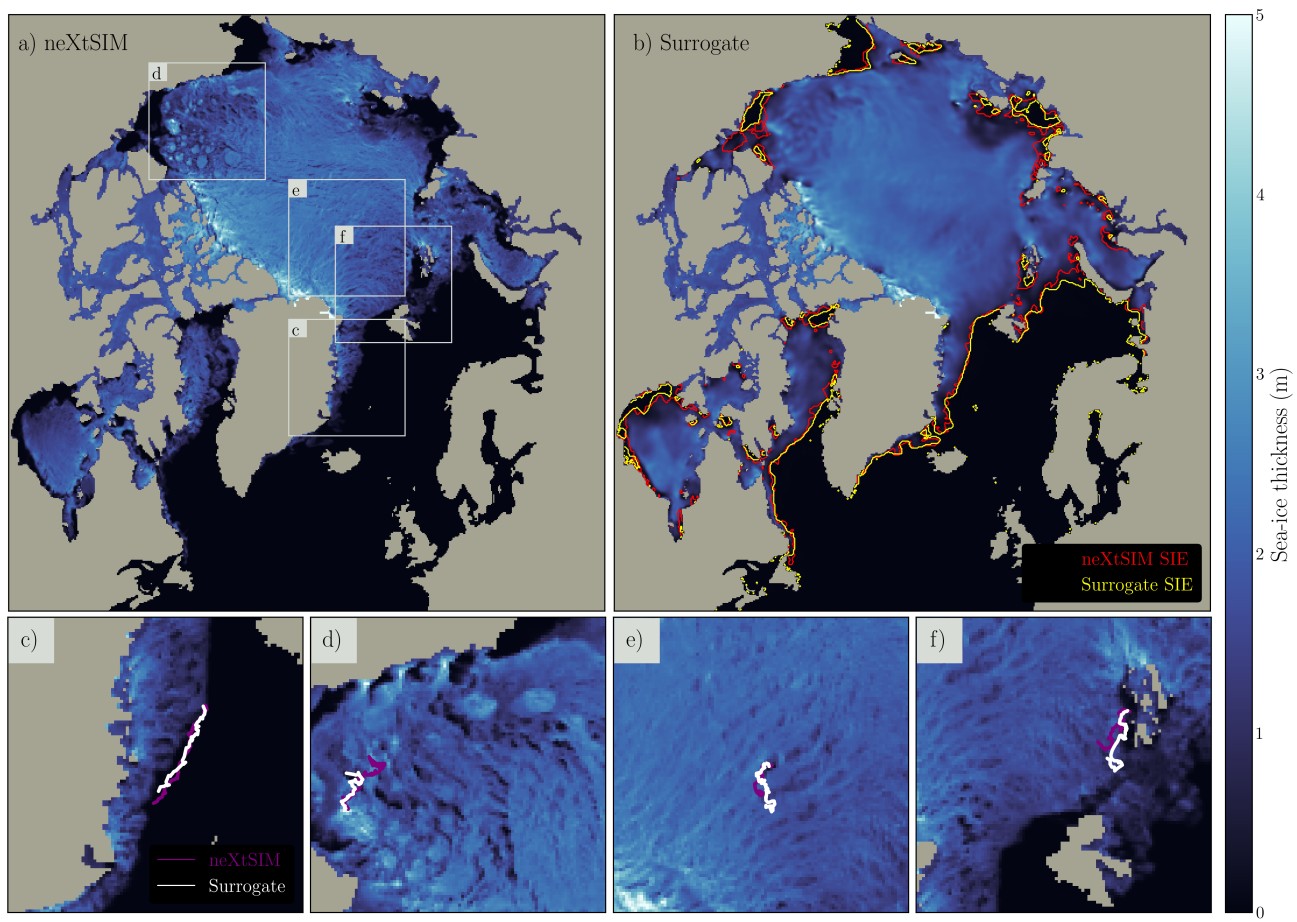

**Figure 6.** Evaluation of the advection from the *1 input constrained* surrogate model. (a) neXtSIM sea-ice thickness (SIT) on May 30, 2018, with zoomed regions indicated for panels c), d), e), and f). (b) Surrogate model output on May 30, 2018, after a 30-day forecast initialized on May 15, with sea-ice extent (SIE) computed from neXtSIM (red) and the surrogate (yellow). Similar delimitation of sea-ice edges is observed in both curves. Panels c), d), e), and f) depict manual feature tracking in different Arctic regions: c) MIZ in Greenland Sea, d) Beaufort Sea, e) Central Arctic, and f) Barents Sea. The trajectories for 30 days are shown in purple for neXtSIM and white for the surrogate model.

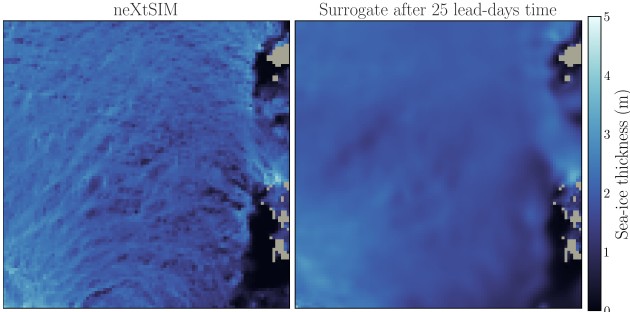

**Figure 7.** Comparison of neXtSIM (a) and surrogate (b) runs for a 25-day period in the Central Arctic. The fine-scale dynamics observed in panel (a) are smoothed in the surrogate model (b).

exponent to be a good quantitative measurement of the diffusion process of the surrogate model. As the number of iteration of the surrogate model grows, we observe that the smoothness of the field increases. This means that we lose information at high frequencies, thus the PSD decreases for high frequencies, see Fig. 8 (b). This reduction for high frequencies further supports the notion that the diffusion processes within the surrogate model leads to a loss of detailed information and finer-scale features. When computing $\beta$, we see a fast increase of $\beta$ when the number of cycle increase, see Fig. 8 (c). In 10 lead-day time, we have an increase of the $\beta$ exponent of 6% averaged over the full year. Interestingly, the spectral exponent does stabilize after 20 lead days and then slowly decrease. We hypothesize that the neural network has reached its highest possible resolution for a correct representation of sea-ice advection at global scale by reducing the inherently chaotic and stochastic fine-scale dynamics. In other words, the surrogate model is able to correctly advect sea-ice thickness up to a given resolution, beyond which smoothing the fine-scale dynamics yields lower RMSEs.

### 5.4 Long-term forecast

In this section, we discuss the ability of the surrogate model to forecast the dynamics of sea-ice thickness at a seasonal scale, focusing on the constrained 1 input surrogate model. Seasonal forecast of Arctic sea-ice is complex (Sigmond et al., 2013), and even more so on a high-resolution grid. While previous results were presented with at most 60 iterations of the surrogate model, we present here runs of the surrogate with 720 iterations which correspond to 360 day forecasts of sea-ice thickness. Those forecasts are initialized from January 2006 to January 2008, with initial conditions sampled every 30 days. In total, we have 25 runs of 360 days to evaluate the surrogate model, see Fig. 9. In Appendix D2, we show some snapshots of the seasonal forecast model over the full year, accompanied by the SIE delimitation, see Fig. D4. The surrogate model is stable over the full year, and the RMSE is lower than that of the daily climatology for 6 months. In the bottom panel of Fig. 9, the global averaged SIT for the surrogate model aligns with both the neXtSIM output and the climatology-based approach. The non-negligible bias of the climatology come from the fact that the daily climatology is computed over $2009 - 2016$ and is

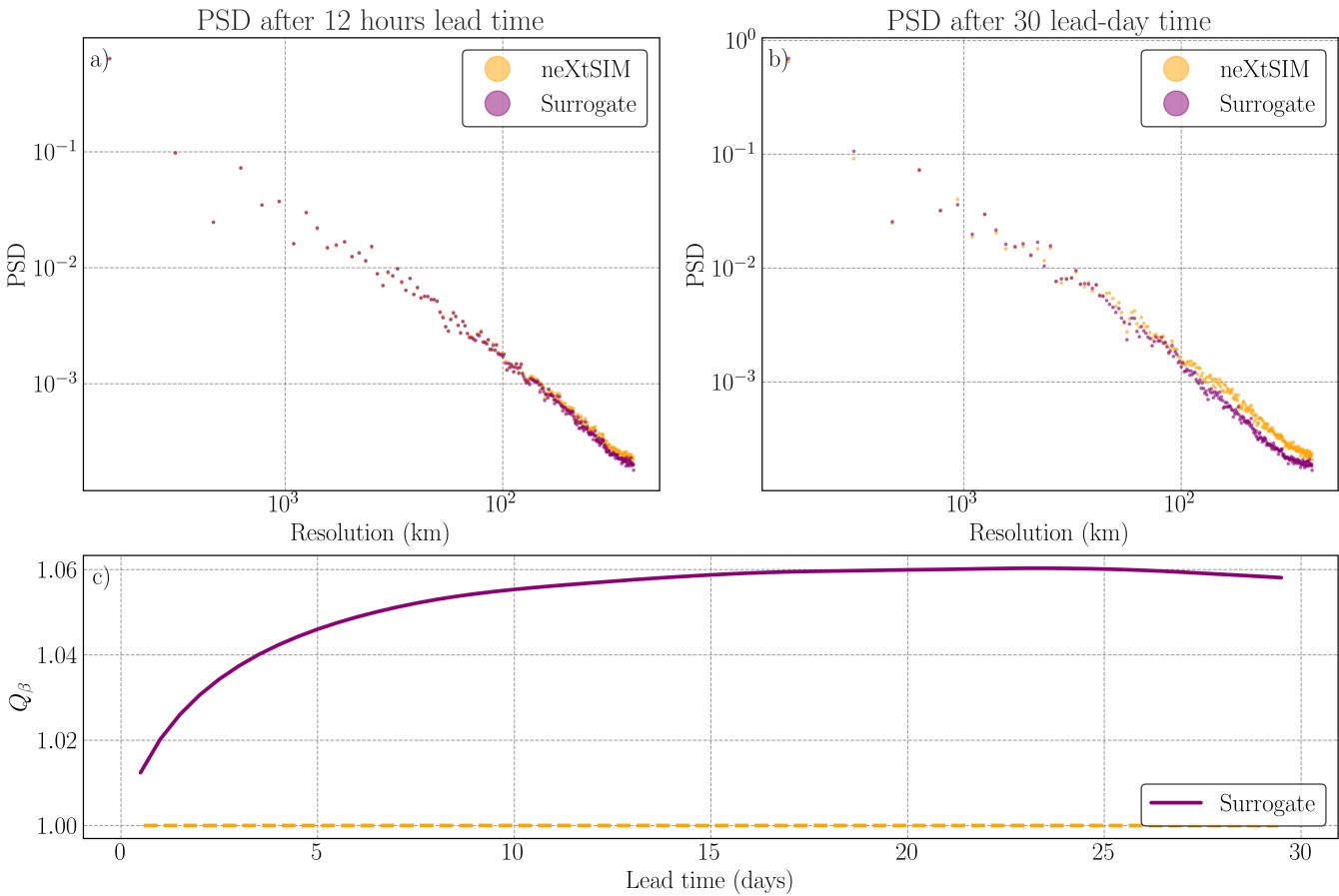

**Figure 8.** Evaluation of power spectral density (PSD) and diffusion process in surrogate modeling. (a) PSD of one neXtSIM output (orange points) and one surrogate output after 12 hours (purple points), indicating a close match. (b) PSD after 30 lead-day time of the surrogate model compared to neXtSIM at the same time, showing lower values for high spatial frequencies in the surrogate model. (c) $Q_\beta$ values quantifying the diffusion process for the surrogate, with the orange dashed line representing neXtSIM for comparison.

directly linked to sea-ice thinning. This consistency can be attributed to the low bias exhibited by the constrained surrogate model, see Fig. 3. The constrained model's low bias helps maintain a physically realistic behavior of the sea-ice, even during long-term forecasts. The low error values of the bias observed during each iteration contribute to maintaining the physical integrity and conservation of the sea-ice in the surrogate model. This indicates that the surrogate model's predictions remain consistent with the overall dynamics of sea-ice, supporting its ability to capture the essential characteristics and behavior of the system. However, it is worth noting that when testing the surrogate model on a two-year forecast, difficulties were encountered in accurately predicting the dynamics beyond one year due to the constant, albeit slow, increase of the error of the surrogate.

In terms of sea-ice extent (SIE) prediction, the surrogate model demonstrates the capability to accurately forecast the edge of the sea-ice throughout the year, regardless of the initialization period, as shown in Figure 10. As anticipated, the surrogate model performs significantly better than persistence during periods of high variation, particularly during summer and autumn. In periods where the ice edge is almost static, the persistence is a good baseline prediction for the position of the ice edge. While our surrogate model still correctly advects SIT, the results for the SIE accuracy do not differ much from those of the persistence. Indeed, since the SIE is post-processed from SIT, and only partially relies on the position of the MIZ, we lose most of the benefit of a correct SIT prediction by the surrogate model. Even though it is a useful marker for the marginal ice zone (MIZ), this post-processed variable is inadequate to represent large scale dynamics of SIT, e.g., in the Central Arctic. It only compares the position of the ice edge, and removes the information about global motion of sea-ice thickness inside the ice sheet.

## 6 Discussion

### 6.1 Fast emulation of high-resolution SIT

Our proposed surrogate model based on a UNet neural network can emulate the large-scale sea-ice thickness as simulated by neXtSIM on daily and seasonal timescales. The main advantage of the emulator is the computational time needed for a forecast. Once the neural network is trained, computing one iteration of the surrogate, a $12\,\mathrm{h}$ forecast, takes approximately $72\,\mathrm{ms}$ on a single NVIDIA A100 GPU. A forecast for one year takes under $1\,\mathrm{min}$. This opens the perspective to run a large ensemble of simulations for complex sea-ice models, which could facilitate data assimilation. Yet, the observed smoothing effects might cause a collapse of the ensemble for longer lead times. This would require further analysis or a subsequent improvement of the surrogate model.

Note that the training of the neural network remains slightly costly, around $18\,\mathrm{h}$ on a single NVIDIA A100 GPU. Approximately, $10000\,\mathrm{h}$ on NVIDIA A100 GPUs were necessary to conduct this study.

Training a surrogate model for a coarser resolution is faster, however, a surrogate model for high resolutions can resolve more processes. To showcase this, we display results for additional experiments on a coarse-grained dataset with $128 \times 128$ grid cells compared to $512 \times 512$ grid cells at the high resolution. The neural network has the same configuration as for the high resolution dataset, and follows the same training procedure. To compare the surrogate trained on the high-resolution

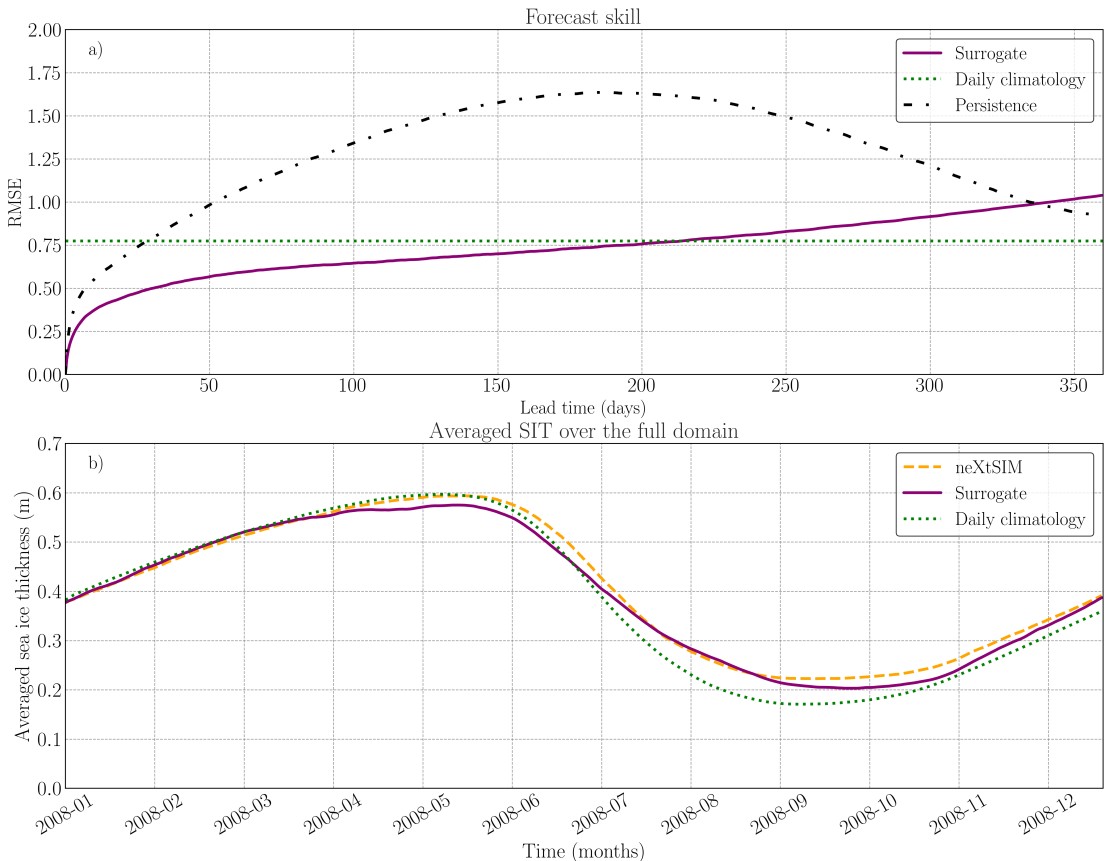

**Figure 9.** Performance evaluation of constrained surrogate model and comparison with persistence and climatology for seasonal forecast. (a) Forecast skill of the constrained surrogate model for year-long forecasts, based on 25 runs with different initialization times. The surrogate model (solid purple) is compared against persistence (dashed black) and daily climatology (dotted green). (b) Global averaged sea-ice thickness throughout the year, starting in January 2008, comparing the surrogate model (solid purple), the physical model neXtSIM (dashed orange), and the daily climatology (dotted green). The seasonality of the SIT is well-preserved by the surrogate.

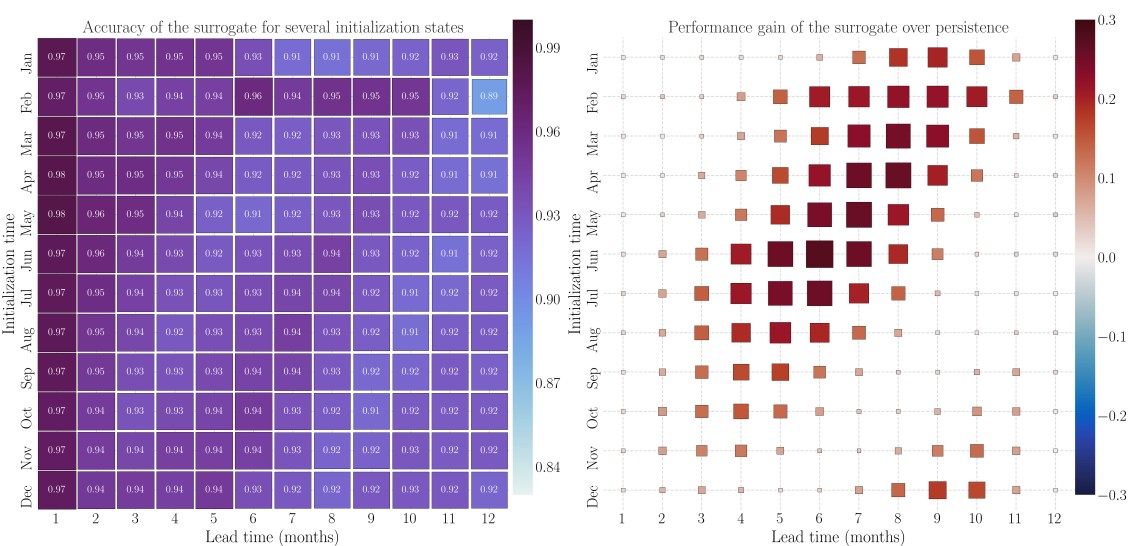

**Figure 10.** Estimation of sea-ice extent forecasting performance for different initialization times and forecast horizons. The left panel illustrates the sea-ice extent (SIE) accuracy on the $2006 - 2008$ test dataset with varying initialization times. The right panel displays the relative difference between the SIE accuracy of the surrogate model and persistence.

dataset to the surrogate trained on the aforementioned coarse resolution, we plot the root-mean-squared error (RMSE) for both resolutions in Fig. D1. Surrogate modeling at the high-resolution decreases the RMSE by $31\%$ after 12 hours compared to the coarse surrogate. This improvement similarly holds throughout resolution levels and also for longer lead times. Since this improvement is visible at all resolutions, it can be linked to a better representation of the small-scale dynamics and their impacts on larger scales for the high-resolution surrogate. While the training and inference of this high-resolution surrogate model are significantly more time-consuming than the coarse-grained surrogate, it is still about 100 times faster than physical model simulations. Consequently, we see a gain by using high-resolution data for surrogate modeling, even if the surrogate leads to smoothed out forecasts.

## 6.2 Architecture of the neural network

Several neural networks have been investigated on a lower resolution set-up, by coarse-graining the dataset down to a $128 \times 128$ grid-cells arrays. On this lower resolution setup, both ResNet (He et al., 2016) and ConvLSTM (Shi et al., 2015) architectures have been studied, yielding quite similar results in terms of forecast skills on the validation dataset. From our experiments, the

specific architecture of the convolutional neural network does not matter much. The LSTM-based approach with a lag of $48\,\mathrm{h}$ led to satisfactory forecast skills; but, because of the high training costs ($441\,\mathrm{s}$/epoch) compared to the UNet ($108\,\mathrm{s}$/epoch), we chose to focus on the UNet structure for the high-resolution dataset. Regarding the ResNet architecture, also implemented with partial convolution, the results were also quite similar, despite higher training costs ($172\,\mathrm{s}$/epoch).

## 6.3 Smoothness of the fields and deterministic neural networks

Using the power spectral density as a quantification of the diffusion shows that the diffusion process reaches a threshold. By omitting the inherently chaotic fine-scale dynamics, which exhibit a stochastic behavior, the surrogate model achieves a balance between capturing the essential large-scale patterns and minimizing the impact of unpredictable fluctuations: chaotic and stochastic processes lead to a high sensitivity to initial conditions, making these processes difficult to model accurately. By prioritizing the larger-scale dynamics and averaging out fine-scale features, the surrogate model mitigates the influence of these chaotic and stochastic processes. This mitigation results into more stable and reliable predictions on a global scale. This hypothesis implies that the surrogate model focuses on capturing the dominant advection patterns that drive the overall behavior of sea ice, while sacrificing some of the finer details. Whereas this trade-off may result in a loss of information for fine-scale dynamics, it allows the model to provide valuable insights into global-scale advection patterns.

The smoothing effects are directly linked to the training of the neural network as a deterministic surrogate model. We can expect less smoothing for better models as the uncertainty is decreased. However, caused by the availability of the training data and by computational considerations, we focus on predicting lead times of $12\,\mathrm{h}$, and there are always situations that cannot be predicted from the available data. By maintaining a deterministic neural network, there will consequently be a smoothing effect, and we generally do not anticipate significant improvements in the quality of details. By adapting generative neural networks, however, the surrogate model could learn to properly sample from the forecast distribution and hence increase the level of fine-scale details (Ravuri et al., 2021). It is worth noting that generative models and especially denoising diffusion models are currently under investigation by different teams for diverse geoscientific problems (e.g., Finn et al., 2023a; Leinonen et al., 2023; Mardani et al., 2023; Price et al., 2023). Nonetheless, the training of such models is notoriously more difficult than that of deterministic models.

Smooth fields are good in terms of RMSE. Caused by the discrete-continuous sea-ice processes, the RMSE might not be an optimal evaluation metric. Training for other metrics can become increasingly more complicated and would exceed the scope of the study. Furthermore, the surrogate model is statistical driven, whereas models like neXtSIM are based on our physical understanding. Such models based on physical principles can have advantages, especially for futures cases where we have an extrapolation task caused by climate change.

## 6.4 Seasonal forecasts

Regarding seasonal forecasts, the model is stable for lead times up to one year, even if the climatology has a smaller forecast error after 6 months. Our surrogate can conserve the average sea-ice thickness over a full year, while also represent advection. This indicates that our surrogate model can capture the large-scale dynamical and thermodynamical evolution of the sea ice

over the full year. These phenomenons are driven by external forcings from the atmosphere and ocean. Yet, the surrogate model can represent the influence of the forcing on SIT, something that a climatology and, especially, a persistence forecast cannot exhibit. Furthermore, for short-term forecasts, the surrogate model consistently outperforms persistence and the daily climatology, and it shows better forecasts than a daily climatology for more than 6 months in terms of forecast skill.

## 6.5 Towards data assimilation and multivariate emulation

The implementation of the surrogate model as neural network allows us to easily compute its adjoint. Thus, we could use a four-dimensional variational (4D-Var) data assimilation scheme. Our current 4D-Var approach under investigation uses the surrogate model primarily for short-term forecasting. Despite the smoothing effect, we believe that the utilization of the adjoint could prove beneficial.

The definition of an adjoint is meaningful for the sea-ice thickness on the projected grid. While performing variational data assimilation on this grid poses no issues, it is important to acknowledge that the constructed adjoint would differ from the one of neXtSIM on the original triangular mesh. However, a common approach in operational variational data assimilation is to apply inner and outer loops (e.g., Rabier et al., 2000). In inner loops, cheaper surrogate models, e.g., the model at a lower resolution, are used, whereas the full high-resolution model is only run in a few outer loops. This could be implemented by applying our neural network surrogate for inner loops and the full neXtSIM model in outer loops.

This study only focuses on predicting the sea-ice thickness, an important variable, especially in operating forecast. However, other variables, like the sea-ice velocity components, as prognostic variables, could be predicted at the same time. Using the interactions between different variables can provide valuable information to the surrogate model. Hence, learning to emulate these variables has the potential to improve the prediction of the sea-ice thickness. Nevertheless, multivariate modeling is a more complex objective for the neural network than univariate modeling. Yet, a successful multivariate surrogate model might offer new perspectives for multivariate variational data assimilation.

## 6.6 Influence of the forcing fields

In this study, we use ERA5 reanalysis forcings, the same forcing product that has forced neXtSIM in our used simulations. Preliminary results of running the surrogate model with forcings derived from CMIP6 model output show that the surrogate can be run with other forcings, Fig. D5. Since this forcing data is based on a free-running simulation, it results into a different evolution of the sea-ice thickness than with the ERA5 reanalysis product. Yet, the model is stable even for others types of forcing. Therefore, the surrogate model has learned to represent the large-scale sea-ice dynamics needed to simulate the sea-ice thickness on daily and seasonal timescales.

## 7 Conclusions

A neural network can emulate the sea-ice thickness at a resolution of $10\,\mathrm{km}$ as simulated by neXtSIM. Trained for prediction of a $12\,\mathrm{h}$ lead time, the neural network can be iteratively applied for surrogate modeling to forecast the thickness for up to one

year. The advantage of the surrogate model over a persistence forecast prevails from the daily timescales, with improvements of around $36\,\%$, to seasonal time scales with more than $50\,\%$ improvement.

We introduce a regularization method for the training of the neural network, constraining the deviations of the global averaged sea-ice thickness from the targeted simulations. This regularization reduces the bias of the neural network and increases the global consistency. The increased consistency then results into a decreased forecast error on daily to weekly timescales.

By adding atmospheric forcings, the surrogate model can represent advective and thermodynamical processes that influence the sea-ice thickness on a large, Arctic-wide, scale. Hence, the seasonal predictions with the surrogate have a predictabilty of 520 up to $8$ months, measured by comparison to the daily climatology.

When the surrogate model is iterated, it exhibits diffusive processes, which lead to a smoothing of the prediction. The predictions are smoothed during the first iterations, as shown by a power spectral density analysis. Whereas the smoothing induces a loss of fine-scale features, it allows the model to stay coherent for the large-scale dynamics that impact the sea-ice thickness. Thanks to this coherency, the surrogate model correctly manages to estimate the global amount of sea ice over the 525 full Arctic. Consequently, the surrogate model can offer a stable low-resolution adjoint for the sea-ice thickness in neXtSIM, for example useable in a variational data assimilation framework.

The surrogate model can make year-long forecasts within a minute. Therefore, the surrogate model presents itself as an opportunity to estimate a large ensemble of simulations. Such a large ensemble can enable the assimilation of previously unused observations into the sea-ice thickness.

## Appendix A: neXtSIM configuration

In this Appendix, we describe more precisely the configuration of neXtSIM used. The model relies for its rheology on the brittle Bingham-Maxwell (Ólason et al., 2022), which is an improvement from the Maxwell-Elasto-Brittle (MEB) rheology (Dansereau et al., 2016). The model equations are solved on an adaptive Lagrangian triangular mesh (Rampal et al., 2016) using a finite element method with a re-meshing protocol. This method helps preserve the gradients in the sea-ice fields which can come from the formations of leads and ridge. The main parameters used for the model are presented in table A1. neXtSIM in this case is coupled with an ocean model (NEMO).

**Table A1.** neXtSIM main parameters, see (Boutin et al., 2023) for more details about the model coupling.

| Parameter | Symbol | Value |
|---|---|---|
| Ice-atmosphere drag coefficient | $C_a$ | $1.6 \times 10^{-3}$ |
| Ice-ocean drag coefficient | $C_w$ | $6.7 \times 10^{-3}$ |
| Scaling parameter for the ridging threshold | $P$ | $3\,\mathrm{kPa/m}^{3/2}$ |
| Main model time step | $\Delta t_m$ | $450\,\mathrm{s}$ |
| Time step for sea-ice dynamics solver | $\Delta t$ | $6\,\mathrm{s}$ |
| Maximum thickness of newly formed ice | $h_{max}$ | $18\,\mathrm{cm}$ |
| Sea-ice albedo | $a_{ice}$ | $0.57$ |
| Snow albedo | $a_{snow}$ | $0.8$ |
| Critical thickness parameter for ice grounding | $k_1$ | $5$ |

## Appendix B: Definition of the accuracy

In order to build an accuracy metric to evaluate the ability of the surrogate to predict the sea-ice thickness, it is necessary to define a threshold value to differentiate non sea-ice grid points from sea-ice grid points. Sea-ice experts commonly define the Marginal Ice Zone (MIZ) with sea-ice concentration between $0.15$ and $0.8$ (Strong, 2012; Comiso, 2006; Rolph et al., 2020). On one month of neXtSIM output (124 snapshots), we compute the grid-points included in this MIZ definition, and then we compute the cumulative distribution of SIT on those grid-points, see Fig. B1. We select a value of $\sigma_{\mathrm{acc}} = 0.1$ for the threshold on SIT to define sea-ice extent. If the grid-point has a SIT above $\sigma_{\mathrm{acc}}$, it is considered a sea-ice pixel, otherwise it is considered either open sea or land.

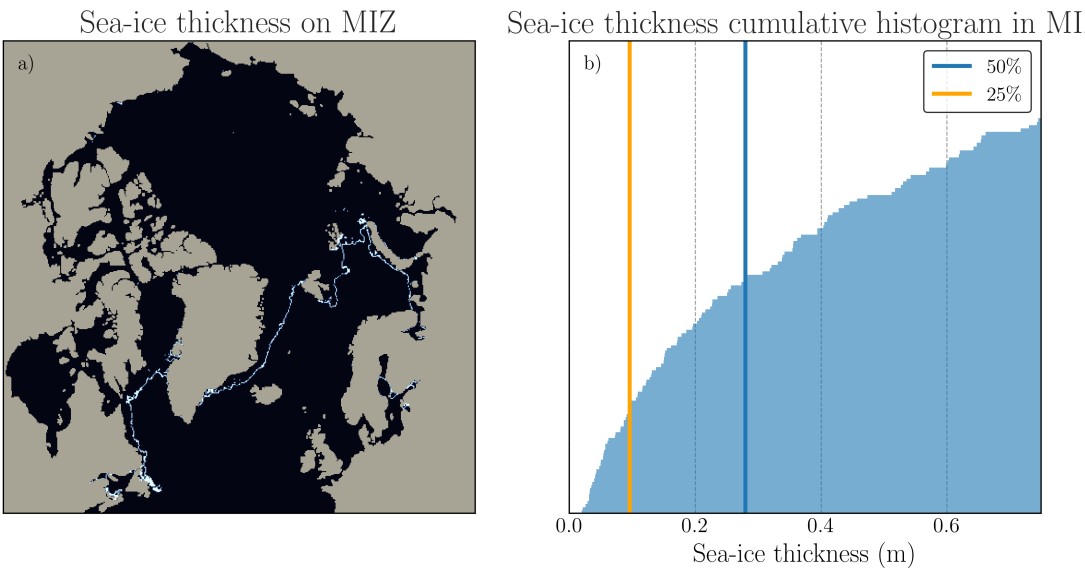

**Figure B1.** Choice of the SIT thickness threshold. On the left panel is shown the SIT thickness on the MIZ commonly defined with SIC. On the right panel is shown the cumulative distribution of this MIZ SIT. The blue vertical line outlines the mean of this distribution and the red line the 25% percentile which coincide with our chosen threshold for the SIT to define the SIE.

## Appendix C: Neural networks architecture

We detail in this section the structure of the neural networks used in the paper. The models are implemented using Tensorflow and Keras. The next sections describe the implementation of partial convolution, and detailed structure of the UNet neural network.

### C1    Partial Convolution algorithm

Instead of the (Liu et al., 2020; Kadow et al., 2020) implementation of the partial convolution where the masks are convoluted alongside the images, we want to keep the mask constant only to represent the land around the sea-ice. Let us define M the mask for which 0 means a land pixel and 1 a valid pixel representing either ice or sea. Our implementation of partial convolution is reported in Algorithm C1.

**Algorithm C1** Partial Convolution pseudocode, it takes as input a tensor of size $(n_b, n_x, n_y, n_c)$, with $n_b$ the batch size, $n_x$ and $n_y$ the image size and $n_c$ the number of channels. It also requires a mask M of size $(n_x, n_y)$, the kernel size of the convolution $k_s$, an $\epsilon$ hereafter set to $10^{-8}$, and an activation function $\sigma$.

---

**Require:** $X, M, k_s, \epsilon, \sigma$

Compute image kernel

Compute $n_p$ the zero-padding around $(n_x, n_y)$

$\tilde{M} = M + \text{padding}$

$\tilde{X} = X + \text{padding}$

Compute $W = \#k_s$

Compute $V = \sum[M = 1]$

Compute $r_m = \frac{W \times M}{V + \epsilon}$

$\tilde{X} = \text{Conv2D}(\tilde{X} + \tilde{M})$

$\tilde{X} = \sigma(r_m X + b)$

**return** $\tilde{X}$

---

## C2 UNet

The detailed structure of the neural network is described in table C1. It has 3 levels of depth, which means that the fields are coarse-grained down to $(128 \times 128)$ pixels, and a total number of $2.4 \times 10^6$ parameters.

## C3 Neural Network training

### C3.1 Losses during training

The losses as described in Section 3 are shown in Fig. C1 for the neural network with 1 input. The validation losses are plotted
in transparency for each associated training losses. We do not observe overfitting which validate the size of the UNet with regard to the size of the dataset. The training of one UNet takes 18 hours on a single Nvidia A100 GPU.

### C3.2 Tuning of $\lambda$

In this section, we describe how the value for $\lambda$ was selected. The value was chosen out of several experiments for different values of $\lambda : 1, 10, 100,$ and $1000$. After training the surrogate models with those values of $\lambda$, an evaluation based on the bias
and the forecast skill (see Sec. 4.2) was done on the validation dataset. The impact of $\lambda$ on the forecast skill was negligible, and was important on the bias. Selecting a value for $\lambda$ that is excessively large could result in a loss of information at the fine scale. A value of $\lambda = 100$ seems to keep a good balance between fine-scale dynamics and global sea-ice thickness.

| Stage | Layer | # params | $n_x$ | $n_y$ | $n_{channels}$ |
|---|---|---|---|---|---|
| Input | PConv | 3776 | 512 | 512 | 32 |
| Down 1 | PConv | 9248 | 512 | 512 | 32 |
| | PConv | 9248 | 512 | 512 | 32 |
| | PConv | 9248 | 512 | 512 | 32 |
| | BatchNormalization | 128 | 512 | 512 | 32 |
| | MaxPooling2D | 0 | 256 | 256 | 32 |
| Down 2 | PConv | 18496 | 256 | 256 | 64 |
| | PConv | 36928 | 256 | 256 | 64 |
| | PConv | 36928 | 256 | 256 | 64 |
| | BatchNormalization | 256 | 256 | 256 | 64 |
| | MaxPooling2D | 0 | 128 | 128 | 64 |
| Bottleneck | PConv | 14771 | 128 | 128 | 256 |
| | PConv | 59008 | 128 | 128 | 256 |
| | PConv | 59008 | 128 | 128 | 256 |
| | PConv | 59008 | 128 | 128 | 256 |
| | BatchNormalization | 1024 | 128 | 128 | 256 |
| Up 2 | UpSampling2D | 0 | 256 | 256 | 256 |
| | PConv | 14752 | 256 | 256 | 64 |
| | Concatenate | 0 | 256 | 256 | 96 |
| | PConv | 55360 | 256 | 256 | 64 |
| | PConv | 36928 | 256 | 256 | 64 |
| | BatchNormalization | 256 | 256 | 256 | 64 |
| Up 1 | UpSampling2D | 0 | 512 | 512 | 64 |
| | PConv | 18464 | 512 | 512 | 32 |
| | Concatenate | 0 | 512 | 512 | 64 |
| | PConv | 18464 | 512 | 512 | 32 |
| | PConv | 9248 | 512 | 512 | 32 |
| Output | PConv | 9248 | 512 | 512 | 32 |
| | BatchNormalization | 128 | 512 | 512 | 32 |
| | PConv | 33 | 512 | 512 | 1 |

**Table C1.** UNet architecture

## Appendix D: Surrogate modeling

In this section, we present more snapshots of the surrogate model predictions, results with different type of forcings and
570 resolution.

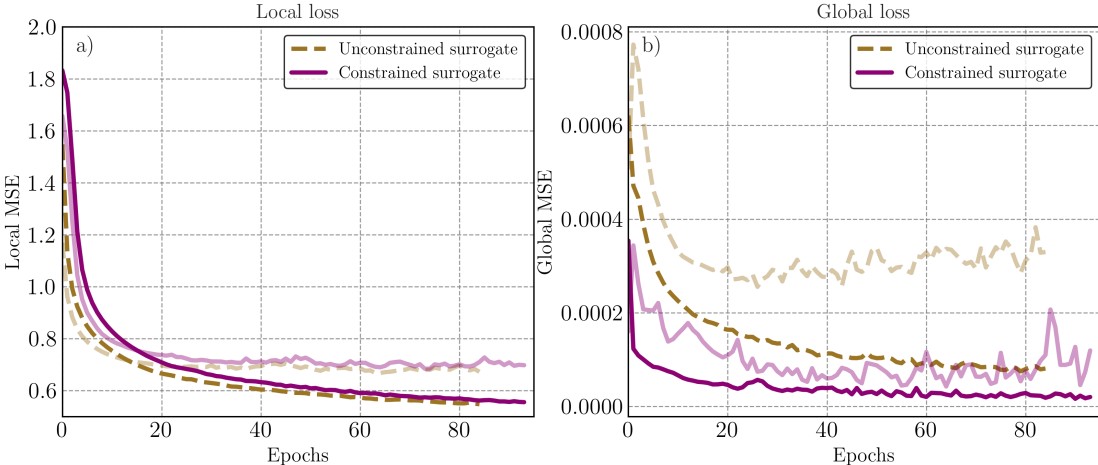

**Figure C1.** Training and validation (in transparency) losses for the neural networks with 1 timestep in the input. Yellow line represent the unconstrained neural network and purple the constrained neural network. We observe that adding the global term in the loss during training allows an important decrease of the global RMSE.

### D1 Impact of the resolution

While we observe some smoothing, we wonder about the gain in training a surrogate model on such a high-resolution model output from neXtSIM since the fine-scale dynamics in the data are smoothed by the surrogate model. Experiments were conducted on a smaller resolution grid, by coarse-graining the original dataset down to $128 \times 128$ pixels. The surrogate model
is trained on this coarse dataset with the exact same method as previously described. The RMSEs from several resolutions are plotted in Fig. D1. It shows a systematic improvement over all resolutions of the surrogate model trained on the high resolution compared to the coarse surrogate model, for different lead times.

### D2 Short-term forecast

In Fig. D2 and D3 we display several snapshots obtained from the constrained surrogate model alongside its corresponding
neXtSIM state.

### D3 Seasonal forecast

In Fig. D4 we can observe several snapshots obtained from the stable constrained surrogate model alongside its corresponding neXtSIM state. For every timestep shown, the surrogate model correctly manages to estimate the global state of the system. Yet, the smooth advection appears at the leading process. It is nonetheless a good approximation of the sea-ice structure during
the full year.

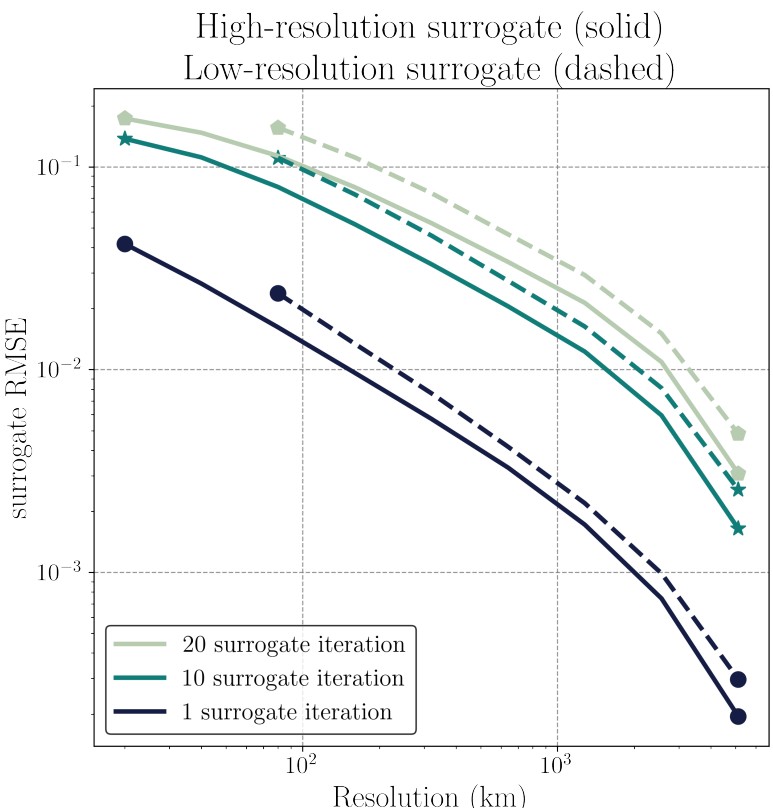

**Figure D1.** Evaluation of the surrogate model RMSE at various resolutions. Two surrogate models trained on high resolution (solid line) and coarse-grained resolution (dashed line) are assessed in terms of RMSE, in comparison to neXtSIM, at different scales and lead times: after 1 iteration (round markers), after 10 iterations (star markers), and after 20 iterations (hexagonal markers).

### D4   CMIP Forcings

As explained in Sect. 6, we evaluate our surrogate model approach with another type of forcings. The forcings are taken from the ECMWF-IFS-HR (25 km atmosphere and 25 km ocean) climate model (Roberts et al., 2017). The surface temperature and velocities are taken, projected on neXtSIM grid and normalized to be able to be fit in the input of the surrogate model. The results are presented in Fig. D5. We see that there is after 50 days a constant bias differences between the surrogate with the different forcings, with the same global behavior. The surrogate model with CMIP6 forcings correctly handle the decrease of the SIE during September, but it has difficulties to match neXtSIM during the next refreezing period. We hypothesize this is caused by the important bias difference between the forcings, neXtSIM being simulated with ERA5 forcings. In any case, our surrogate model remains stable when changing the forcings.

Surrogate forecast after 5 days lead time

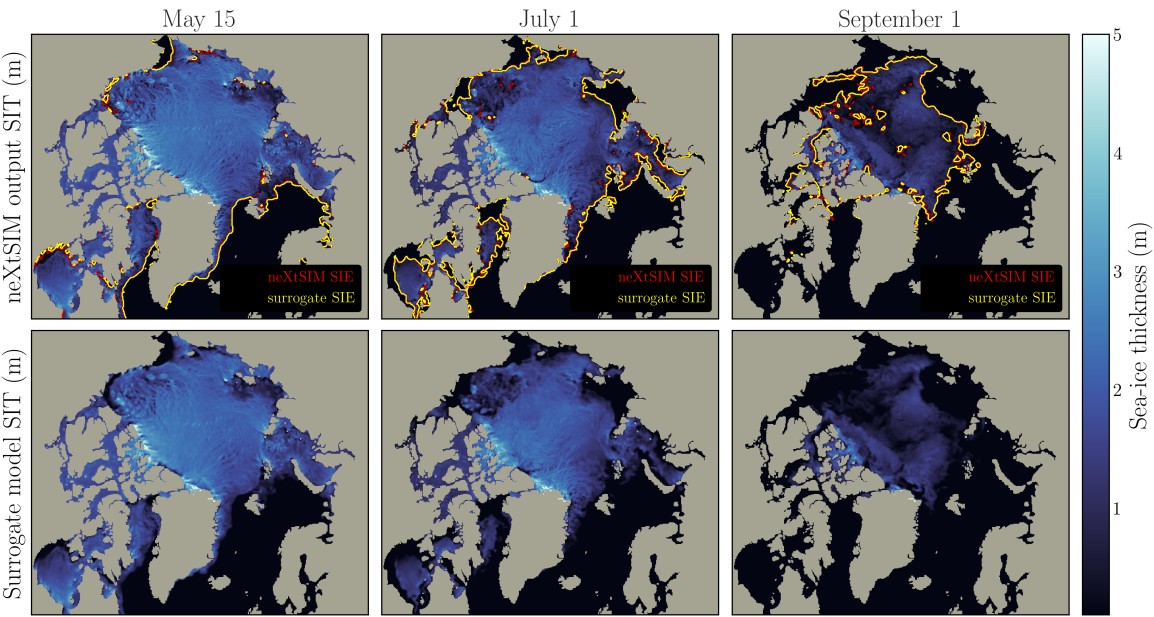

**Figure D2.** The figure presents the surrogate model using three different initialization states (May 15, July 1, and September 1). The surrogate model, with constrained inputs and 1-timestep configuration, is run for 5 lead days. The top panel illustrates the neXtSIM output, while the middle panel showcases the surrogate model output. Contour lines representing the sea-ice extent are displayed on the neXtSIM panel for both neXtSIM (red) and the surrogate (yellow), while the surrogate model examples omit these contours for clarity.

Surrogate forecast after 25 days lead time

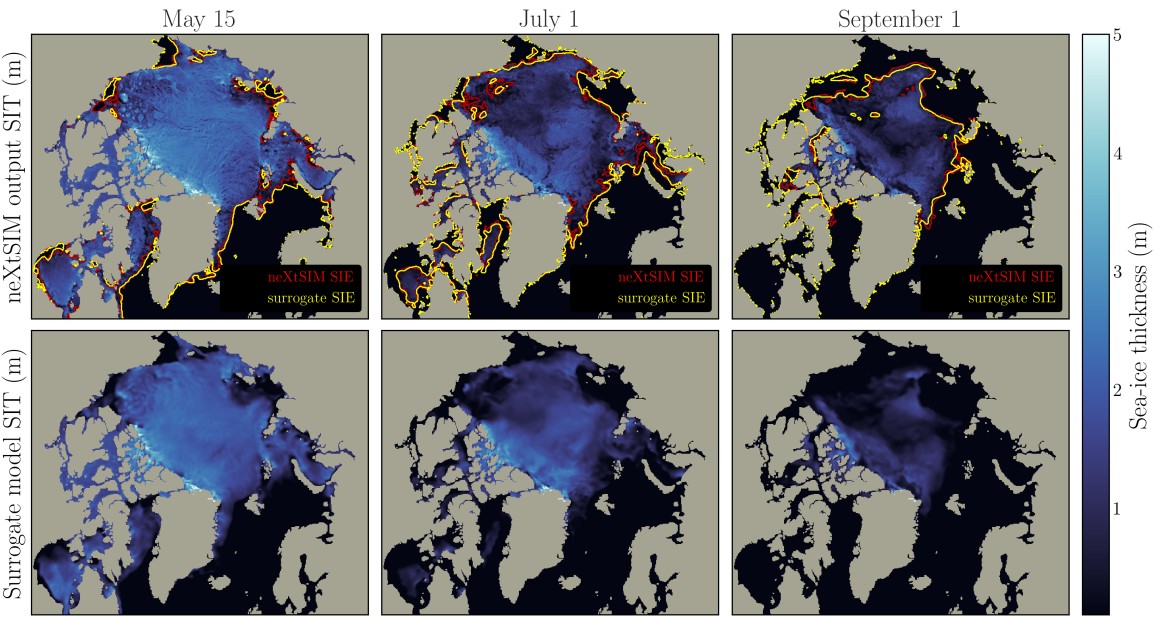

**Figure D3.** The figure presents the surrogate model using three different initialization states (May 15, July 1, and September 1). The surrogate model, with constrained inputs and 1-timestep configuration, is run for 25 lead days. The top panel illustrates the neXtSIM output, while the middle panel showcases the surrogate model output. Contour lines representing the sea-ice extent are displayed on the neXtSIM panel for both neXtSIM (red) and the surrogate (yellow), while the surrogate model examples omit these contours for clarity.

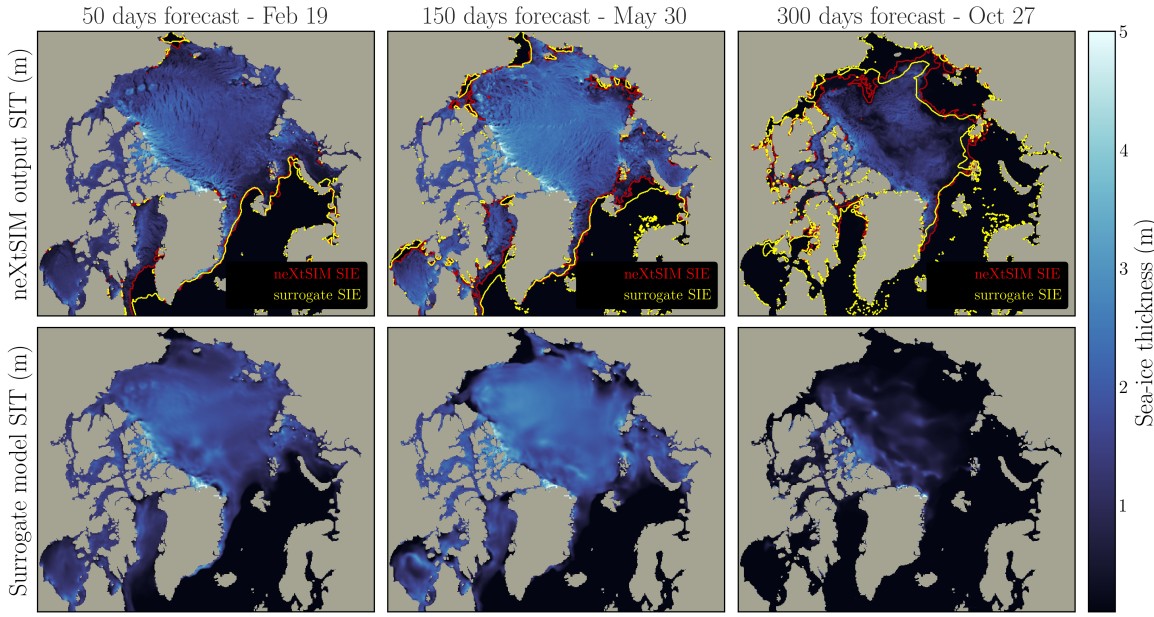

**Figure D4.** Snapshots for seasonal forecast of neXtSIM and the surrogate. The surrogate model is run starting from January 1st 2006, for 720 iterations. Results after 100, 300, and 600 iterations are presented in this figure. Results are plotted with the neXtSIM output above and the surrogate model in the middle panel. On the neXtSIM panel are plotted the contour of the sea-ice extent for both neXtSIM (red) and the surrogate (yellow). For better clarity, those contours are not represented on the surrogate model examples.

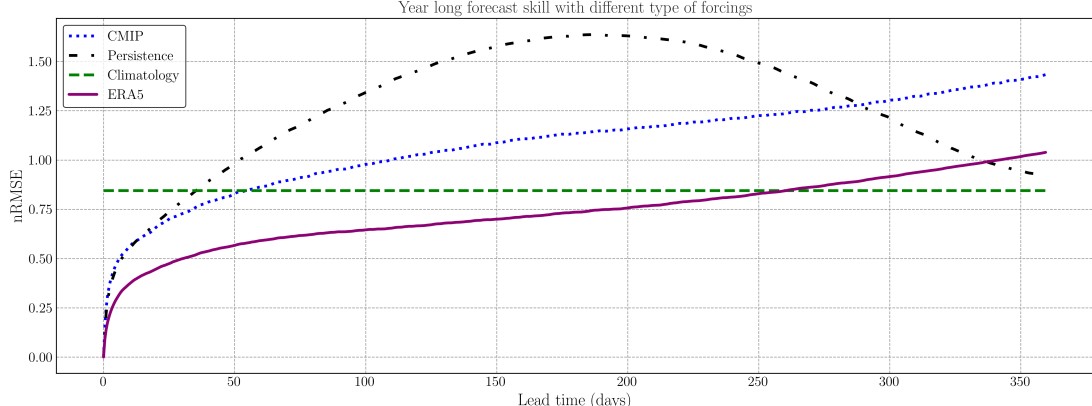

**Figure D5.** Seasonal forecast of the surrogate model while changing the type of forcings in 2018. The neural network remains exactly the same as the trained one. It is thus trained with ERA5 forcings. In the test part, the forcings are changed to CMIP6 forcings (blue dotted line). This forecast skill is compared with neXtSIM and ERA5 forcings (purple solid line), persistence (black dashed line) and daily climatology (green dashed line).

## D5  Quantification of diffusion

In this section, we describe how the spectral exponent of the power-law is computed, following Clauset et al. (2009). After normalizing the data coming from Eq. (15), with Simpson's rule, we fit the distribution with a function of the form

$$p(x) = \frac{\beta - 1}{x_{\min}} \left( \frac{x}{x_{\min}} \right)^{-\beta}, \tag{D1}$$

for $x > x_{\min}$, which is manually chosen. The log-likelihood function becomes

$$\mathcal{L}(\beta) = \log \prod_{i=1}^{n} \frac{\beta - 1}{x_{\min}} \left( \frac{x_i}{x_{\min}} \right)^{-\beta}, \tag{D2}$$

with $x_{i \in (1,\dots,n)}$ the $n$ points above $x_{\min}$. By differentiating this likelihood with respect to $\beta$, and setting the result to $0$, we obtain its maximum, which yields the estimator equation:

$$\hat{\beta} = 1 + n \left[ \sum_{i=1}^{n} \log \frac{x_i}{x_{\min}} \right]^{-1}. \tag{D3}$$

The computation of the spectral exponent is performed using this technique across all samples of the test dataset and all lead times, under the same restrictions. We exclude the first 10 points and the last 20 points of the distribution to focus on the linear part of the PSD. Indeed, as depicted in Fig. 8b, after some time, a flattening of the PSD is observed on the fine scale, directly linked to the smoothing of the emulated sea-ice thickness.

*Code and data availability.* The authors will provide access to the data and weights of the neural networks upon request. The source code for the experiments and the neural networks is publicly available under https://github.com/cerea-daml/nextsim-surrogate. The outputs of neXtSIM model will be made available upon requests. Forcings data are publicly available in the Copernicus Data Store https://cds.climate.copernicus.eu

*Video supplement.* A video of the seasonal forecast for the year 2017 is available at https://doi.org/10.5446/62131

.

*Author contributions.* EO provided the data and the insigths about neXtSIM. GB ran the simulation of neXtSIM and build the SIT dataset. CD, TSF, AF, and MB refined the scientific questions and prepared an analysis strategy. CD performed the experiments. CD, TSF, AF, and MB analyzed and discussed the results. CD wrote the manuscript with TSF, AF, MB, GB, and EO reviewing.

*Competing interests.* The authors declare that they have no conflict of interest.

*Acknowledgements.* The authors acknowledge the support of the project SASIP (grant nr. 353) funded by Schmidt Futures – a philanthropic initiative that seeks to improve societal outcomes through the development of emerging science and technologies. This work was granted access to the HPC resources of IDRIS under the allocations 2021-AD011013069 and 2022-AD011013069R1 made by GENCI. The authors would like to thank Pierre Rampal, Laurent Bertino, Anton Korosov, Julien Brajard and their colleagues from NERSC for their insightful inputs. The authors would also like to thank two reviewers, Nils Hutter and one anonymous reviewer, for their comments and remarks that helped improve the manuscript. CEREA is a member of the Institut Pierre-Simon Laplace (IPSL).

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
