# Peer review of "Data-driven surrogate modeling of high-resolution sea-ice thickness in the Arctic"

_EGUsphere, 2023_

## Author Comment (AC1)

**Response To Referee 1**
**for 'Data-driven surrogate modeling of high-resolution sea-ice thickness in the Arctic'**

Charlotte Durand, Tobias Sebastian Finn, Alban Farchi,
Marc Bocquet, and Einar Òlason

December 2023

**RC: Reviewer Comment**; AR: Author Response

RC: In this manuscript, the authors present a machine-learning based surrogate model of the numerical sea-ice model neXtSIM. The presented surrogate model simulates sea-ice thickness and its predictions outperform the climatology benchmark for lead times up to 8 months. The findings of this study are a valuable addition to the field and illustrate how machine learning can be used to reduce the computational costs of sea-ice simulations, e.g. in the context of ensemble forecasting. The main shortcoming of the surrogate model presented, however, is that it simulates very smooth thickness fields compared to the feature-rich neXtSIM input data used for training that includes for example leads. The authors address and analyze this issue in the manuscript, but I still have the major comments outlined below regarding the presentation, analysis, and interpretation of this point that should be addressed before I can recommend this paper for publication.

AR: We deeply appreciate the reviewer's thorough and insightful review of our work. In the following, we respond to the comments and raised issues and point to the changes in our manuscript.

RC: Major comments:
Smoothness of simulated ice thickness fields:

RC: The authors use simulations of neXtSIM, known for its ability to resolve deformation features and heterogeneous sea ice fields, to train the NN emulator. The surrogate model presented in

this manuscript is not able to retain these features over multiple iterations and quickly smoothes the sea ice fields, resulting in thickness fields much more similar to coarse-resolution sea-ice simulations. The authors argue that the smoothed version better minimizes the RMSE (MSE used for training) compared to a model that retains these features and potentially gets penalized for misplacing them. Therefore, the model learns to predict smooth fields and mimics large-scale circulation. While this makes sense in light of the cost function used, I have the following issues with how this fact is presented and interpreted:

RC: Why do you use high resolution in the first place? In the abstract and introduction, the authors make the valid point that current models that simulate small-scale features, like e.g. leads, are computationally very expensive and that a surrogate model would be of great benefit here. However, the presented surrogate model is not able to simulate these fine-scale features, but "just" the large-scale dynamics. Now, I am wondering why it is necessary to use the feature-rich simulations for training. In the introduction the authors suggest that "small-scale effects have an advantage on representing the thermodynamics of sea ice", but I am not aware of modeling studies that have proven this point comprehensively. Now I am wondering if the same results could also be achieved with a coarse resolution model that also resolves the large-scale circulation (and is much cheaper to run). Once having trained the NN on coarse resolution model output, the authors should comment on whether there is an additional benefit in using the high-resolution input data that is currently used.

AR: Training a surrogate model for a coarser resolution is faster, however, a surrogate model for high resolutions can resolve more processes. To showcase this in the following, we display results for additional experiments on a coarse-grained dataset with $128 \times 128$ grid cells compared to $512 \times 512$ grid cells at the high resolution. The neural network has the same configuration as the high resolution dataset, and follows the same training procedure. To compare the surrogate trained on the high-resolution data to the surrogate trained on the aforementioned coarse resolution, we present the root-mean-squared error (RMSE) depending on the resolution in Fig. 1. The RMSE of the high-resolution surrogate model is decreased by 31% after 12 hours compared to the coarse surrogate. This improvement similarly holds throughout resolution levels and also for longer lead times. Since this improvement results out of different modelled resolutions, it can be linked to a better representation of the small-scale dynamics and their impacts on larger scales for the high-resolution surrogate. While

the training and inference of this high-resolution surrogate model are significantly more time-consuming than the coarse-grained surrogate, it still maintains a rapid processing speed compared to physical model simulations. Consequently, we see a gain by using high-resolution data for surrogate modelling, even if the surrogate leads to smoothed out forecasts. To clarify the use of high-resolution data in our manuscript, we will add Fig. 1 with a brief discussion to the appendix. Additionally, we will add to the discussion (Sect. 6) a paragraph about the use of high-resolution data to emulate neXtSIM sea-ice thickness (SIT).

RC: **Are the smoothed fields really simulating large-scale dynamics and are the presented methods sufficient to show this? The authors compare their NN results against persistence and find increased skill of the surrogate model. They attribute this skill to the fact that the model learned the large-scale dynamics. In Fig. 10 one can see that the model only outperforms persistence forecasts in periods of rapidly changing ice cover and thickness (melt and early freeze period). Couldn't we find the same behavior as well, if the model learned the climatology of ice thickness and relaxes the input sea ice thickness to this climatology? This would be a better benchmark to beat to justify the authors' claim of learned physics. In general, there should be a more in-depth analysis to demonstrate that the model learned large-scale dynamics, for example, how does the integrated ice edge error (Goessling et al., 2016) varies for different lead times (a quantitative analysis of the qualitative comparison in Appendix D), or a more quantitative evaluation of ice drift started in Fig. 6. Another possibility would be to compare the model skill at different spatial scales by e.g. coarse-graining the predictions. Currently, metrics based on pixel values are shown that always include the effect of missing features. In a scale-dependent analysis, the authors could see up to which scale the model has improved skill and if the large-scale variations are represented appropriately. Or can similar information be extracted from your power spectrum analysis?**

AR: Firstly, in Fig. 10 of the original submitted manuscript, the variable under consideration for the evaluation of long-term forecast is an accuracy based on sea-ice extent (SIE). We define the SIE with a SIT threshold of $10\,\mathrm{cm}$ and then define an accuracy. As written in the paper : *By obtaining a classification mask between ice and no ice, we can easily define an accuracy metric based on the SIE. We define two terms: the first one $N_{>\sigma_{\mathrm{acc}}}$ indicates the number of pixels where $\mathbf{x}_{t_n+k\Delta t}$ and $\mathbf{x}^{\mathrm{f}}_{t_n+k\Delta t}$ disagree on the presence of sea ice, and the second one $N_{<\sigma_{\mathrm{acc}}}$ where the models disagree on the presence of open water. The accuracy is averaged over all*

[Figure]

Figure 1: Evaluation of surrogate model RMSE at various resolutions. Two surrogate models trained on high resolution (solid line) and coarse-grained resolution (dashed line) are assessed in terms of RMSE, in comparison to neXtSIM, at different scales and lead times: after 1 iteration (round markers), after 10 iterations (star markers), and after 20 iterations (hexagonal markers).

$N_s$ *samples:*

$$\mathrm{acc}_{\mathrm{SIE}}(k) = \frac{1}{N_s} \sum_{n=1}^{N_s} \left(1 - \frac{N_{>\sigma_{\mathrm{acc}}}(t_n + k\Delta t) + N_{<\sigma_{\mathrm{acc}}}(t_n + k\Delta t)}{N_{\mathrm{sea-ice\,pixels}}}\right). \quad (1)$$

This definition is actually extremely close to the Integrated Ice Edge Error (IIEE), the main difference being that our threshold is based on SIT and not on the sea-ice concentration (SIC). Even though it is a useful marker for the marginal ice zone (MIZ), this post-processed variable is inadequate to represent large scale dynamics of SIT, e.g., in the Central Arctic. It only compares the position of the ice edge, and removes the information about global motion of sea-ice thickness inside the ice sheet.

Secondly, our surrogate can conserve the average sea-ice thickness over a full year, while it also can represent advection. This indicates that our surrogate model can capture the large-scale dynamical and thermodynamical evolution of the sea ice over the full year. These phenomenons are driven by external forcings from the atmosphere and ocean. Yet, the surrogate model can represent the influence of the forcing on SIT, something that a climatology and, especially, a persistence forecast cannot exhibit. Furthermore, for short-term forecasts, the surrogate model consistently outperforms persistence and the daily climatology and it shows better forecasts than a daily climatology for more than 8 months in terms of forecast skill. The IIEE over the full year exhibits different behavior between the climatology and the surrogate model score, as presented in Fig. 2, where the skill of the surrogate is decreased for longer lead times compared to the climatology. If the surrogate model would have only learned to relax the forecast towards the climatology, we would expected the IIEE score to be much closer to the climatology for longer lead times. We will add more comments in the discussion and openings about its implication and to clarify those points.

**RC: If smooth fields are better for prediction, why should the scientific community then at all pursue developing feature-rich models like neXtSIM in a prediction context?**

AR: This is just a first step in surrogate modeling of a rich model like neXtSIM, and we can hope that more advancement would bring more fine-scale features even at longer lead times. Smooth fields are good in terms of RMSE. Caused by the discrete-continuous sea-ice processes, this might not be an optimal evaluation metric. Training for other metrics can become increasingly more complicated and would exceed the scope of the study. Furthermore, the surrogate model is statistical driven, whereas models like neXtSIM are based on our physical understanding. Such models based on physical principles can have an advantage especially for futures cases where we have an extrapolation task, caused by climate change. We will discuss this point in Sect. 6.

[Figure]

Figure 2: Time evolution of the IIEE over 2018 for the surrogate model at different lead time and daily climatology. IIEE scores are based with neXtSIM as the ground truth.

**RC:** **The authors suggest that the surrogate model will be of great advantage in computing the adjoint model in variational data assimilation or generating larger ensemble sizes. I have four comments on this: 1) The surrogate model smoothened the input fields and by doing so will reduce the spread of an ensemble, potentially limiting its use for data assimilation. 2) Given the smoothness of the simulated thickness fields and the strong differences with the feature-rich input fields, do you think the surrogate and the numerical model are similar enough to use the surrogate for the adjoint, especially over longer assimilation windows? 3) Does using the surrogate model as adjoint work that easily given the interpolation from unstructured to regular grid? 4) To properly use the surrogate model in data assimilation for both creating an ensemble or the adjoint, more model variables should be simulated than just sea ice thickness. Please comment on all these points in the manuscript.**

AR: Thank you for your feedback. We appreciate your remarks, and we will incorporate more details on these points in the discussion section. Regarding your first point, it is a valid consideration. Yet, it is important to note that our initial emphasis will be on variational data assimilation, we do not necessarily think of employing ensembles. Note however that most ambitious implemented 4D-Var are in practice based on approximate adjoints, or exact adjoints of a simplified model. Nevertheless, we recognize the relevance of your remark.

On the second point, our current approach involves using the surrogate model primarily for short-term forecasting. Despite the smoothing, we anticipate obtaining meaningful results and believe that the utilization of the adjoint could still prove beneficial.

Concerning the third point, this is a relevant point. The definition of an adjoint is meaningful concerning the sea-ice thickness field on the projected grid. While performing variational data assimilation on this grid poses no issues, it is important to acknowledge that it might involve a different adjoint compared to the original triangular mesh. However, a common approach in operational variational data assimilation is to use inner and outer loops. In inner loops, cheaper surrogate models, e.g. the model at a lower resolution, are used, whereas the full high-resolution model is only run in the outer loops. Compared to the raw minimization of the variational cost function by employing the full model, there is no guarantee of convergence. This methodology is nevertheless used with much success and might also also a way for such hybrid approaches.

On the fourth point, we completely agree. Our upcoming objectives include the training of a multivariate surrogate model, which can be also used for data assimilation.

**RC:** **Is it in general not possible to achieve a higher degree of details in the ice thickness or is it your outlined training and network architecture that hinders it? In the introduction, you state that the cost function plays a major role here, but the manuscript lacks suggestions how to potentially overcome this issue. Please outline potential ways forward in the paper.**

AR: The smoothing effects are highly related to the training of the neural network as deterministic surrogate model. We can expect less smoothing for better models as the uncertainty is decreased. However, caused by the availability of the training data and by computational considerations, we focus on predicting lead times of 12 hours, and there are always futures that cannot be predicted with the available data. By maintaining a deterministic neural network, there will consequently be always a smoothing effect, and we generally do not anticipate significant improvements in the quality of details. By adapting generative neural networks, however, the surrogate model could learn to properly sample from the possible futures and achieve a higher level of fine-scale dynamics. It is worth noting that generative models are currently under investigation by different teams for diverse geoscientific problems Finn et al. (2023). Nonetheless, the training of such models is notoriously more difficult than for deterministic models. As suggested, we will expand on this in the proper place at the discussion part of the manuscript.

**RC:** **Text quality: The manuscript follows a clear structure, but the**

text is in passages hard to read and follow and clearly requires further editing. In parts, words are missing or sentences are half-finished. In times of automated language editing tools, more thorough language editing is possible also for non-native speakers, and I highly encourage the authors to make use of these tools in the future.

RC: L27: "of the Arctic" → Consider removing "of the Arctic" as neither CICE nor SI3 are limited to the Arctic

AR: We will remove those terms from the sentence.

RC: L28: "road" → route?

AR: We will correct this term, it comes from a bad translation from french.

RC: L32: "Divergent features in the ice, like leads and polynyas" → Polynyas are not necessarily formed by divergence.

AR: Thank you for your remark, we will modify the sentence accordingly.

RC: L34-35: "Consequently, models correctly representing the effects of such small-scale can have also an advantage in representing the thermodynamics of sea ice." → Could you please add references, on which studies you base this general statement? In my eyes, it is still an ongoing research question if and what advantage these directly resolved small-scale features have in contrast to parameterizations currently used in climate models.

AR: We agree this is an ongoing research problem and there is no citation available to make this statement, we will modify the sentence in this sense to make it clearer.

RC: L34: "small-scale" → features? Processes? A word seems missing here.

AR: Thank you for your remark, indeed, we meant 'processes' and we will correct the sentence accordingly.

RC: L39: "benefit" → benefits

AR: Thank you for your remark, we will correct the sentence.

**RC: L60:** "Explained differently, the surrogate model is trained to reduce errors" → I do not see how this explains the sentences before differently, it basically says the same as the first sentence in L59. Please clarify.

AR: Thank you for your remark, we will remove this sentence as the first sentence is already clear enough.

**RC: L65:** "learn" → train?

AR: Thank you for your remark, we meant 'train' and we will correct the sentence accordingly.

**RC: L72:** "surrogate model" → simulate?

AR: Thank you for your remark, we meant 'emulate' and we will correct the sentence accordingly.

**RC: L84:** "model area" → It is not clear if this is the area of the neXtSIM simulations or of the NN. Please clarify the text accordingly.

AR: We will clarify the sentence, we meant the simulations area.

**RC: L106:** "Because these forcings are also to guide the neXtSIM simulations" → From this statement I assume that the neXtSIM simulations are also forced with ERA5. Please clarify this already earlier on in the text to prevent confusion.

AR: We will clarify this in the paragraph above, neXtSIM simulations are indeed forced with ERA5.

**RC: L131:** "add to the inputs the SIT" → Didn't you write above that SIT is already an input? Why add it again? Please clarify.

AR: This sentence meant that we also included in the training dataset SIT and forcings at time $t - \Delta t$, we will make this addition more explicit in the sentence.

**RC: L133-134:** "there are called later 'with 2 inputs'. Otherwise, the neural networks are trained 'with 1 input' " → Could you please add those labels for clarity to Table1.

AR: Thank you for your remark, we will add those labels to the Table for clarity

RC: **L152: "(Rampal et al., 2019)" → This reference is somehow misleading at it is not clear that it only refers to the multi scale features in sea-ice dynamics, and not to the ability of CNNs to represent those. Please clarify this, or remove the citation here.**

AR: Thank you for your remark, this citation is indeed misleading, and we will correct the sentence accordingly.

RC: **Figure 2 Caption: "512, 256, and 128," → The figure also shows images of size 64. Please correct.**

AR: Thank you for your remark, there is one additional layer of depth in the figure, we will correct it accordingly.

RC: **L165: "sea" → ocean?**

AR: Thank you for your remark, we meant "ocean" and we will correct the sentence accordingly.

RC: **L188: "global mean of x and y" → As x and y have a physical meaning, it would be helpful for readers if you could also write what the local and global loss mean with respect to sea ice, e.g. local and global trends in sea ice thickness.**

AR: Thank you for your remark, we will add this clarification in the sentence.

RC: **L192: "$\lambda$ is manually tuned to 100" → What do you optimize for, how do you manually decide on best performance? Please clarify.**

AR: Thank you for your remark, we will add the detail of the tuning choices in the manuscript. The choice was made with an evaluation on the validation dataset, for several values of $\lambda$, of the forecast skill and the bias. Non-chosen values for $\lambda$ were not evaluated on the testing dataset. The change in the value of $\lambda$ did not have a significant influence on the forecast skill, except for $\lambda = 1000$, in which case there is an important decrease of the forecast skill. The influence of $\lambda$ was more pronounced on the bias error and has guided our choice towards $\lambda = 100$.

**RC: L224-225:** "over all pixels (i, j) of the field of size (Nx , Ny )" →
Also over land pixels? Including land pixels in the RMSE will
artificially reduce its value.

AR: Yes, we compute the RMSE over all pixels, but since it is also done for
every baseline, this artificial reduction does not change the meaning of the
results. Yet, we will correct this, and modify accordingly all the figures.

**RC: L235-237:** "We define two terms: the first one $N > \sigma$acc indicates
the number of pixels where xtn+k$\Delta$t and xftn+k$\Delta$t disagree on
the presence of sea ice, and the second one $N < \sigma$acc where the
models disagree on the presence of open water." → This is not
clear to me. For all pixels, where the two masks disagree, one
will show ice and the other will show open water. Shouldn't
therefore not also both terms be the same? Please check those
definitions and clarify.

AR: As mentioned above, this is actually really similar to the IIEE definition,
with the overestimation term and the underestimation term. We will
clarify this in the paragraph as it is indeed misleading.

**RC: L250:** "kx and ky" → Please rename the indexes x and y to not
confuse them with the input of the model x and the output y.

AR: Thank you for your remark, we will correct the indices to avoid the con-
fusion.

**RC: L254:** "justified" → caused?

AR: Thank you for your remark, we will change the work 'justifed' for better
clarity.

**RC: L259-260:** "In practice, this exponent can be numerically esti-
mated by a linear regression between lnE and $ln|k|$." → multiple
studies show that linear fits in double logarithmic plots are not
ideal for determining power-law exponents, e.g. Clauset et al.
(2009). Please elaborate why you chose this method. Also how
does you metric takes into account if such a scaling actually ex-
ists or not, or are you computing exponents regardless of the
distribution?

AR: Thank you for your remark; we were not aware of those studies, and chose
this method for its simplicity. We will correct this computation with a

better alternative. We are also computing the regression score for this power law, which were checked, and are systematically high, for the short-term forecast.

**RC: L 270: "in" → on**

AR: Thank you for your remark; we will correct the sentence accordingly.

**RC: L281-282: "However, the impact of the global Eq. 6) on the RMSE relatively small compared to the influence of including additional time steps." → This sentence does not fit to the observed results. When adding the constraint to the 1 input NN the global RMSE reduces by an order of magnitude (as written in the sentence before). Comparing both unconstrained NNs, the global RMSE reduces to about 25% when including additional time steps, so a much lower reduction compared to including the constraint.**

AR: Thank you for your remark. This analysis was related to the RMSE and not the global RMSE, for which it remains true. We will make this more evident in the sentence.

**RC: L. 283: "The impact of adding.." → Just "Adding..."**

AR: Thank you for your remark; we will correct the sentence accordingly.

**RC: L288: "reduce" → the RMSE increases!**

AR: Indeed we meant increase... we will correct this in the manuscript.

**RC: L 289: "surrogate RMSE" → what is the surrogate RMSE?**

AR: It meant the RMSE of the surrogate, but this 'surrogate' work is indeed misleading, we will remove it for better clarity.

**RC: L291-292: "after 12 hours, the global RMSE has improved by a factor 9.4 for the one input surrogate" → Please comment why there is so little improvement for the 2 input NN.**

AR: The global RMSE, with a smaller value than the RMSE, during the training, is much more volatile. While the training takes into account the global loss, and try to reduce it, in the case of the 2 inputs NN, the best

model, according to validation loss (the sum of the local and global losses, weighted by $\lambda$), add a given global RMSE which happened to be similar to the model trained without the constant. The fact that the results are still better in constrained case after 15 days, is also a proof that we indeed learned from the global loss term. We will add these discussion points to the manuscript.

**RC: P 307: "leaded" $\rightarrow$ ??**

AR: We will correct this sentence; we meant 'could have lead'.

**RC: L308: "both biases" $\rightarrow$ which biases?**

AR: We will clarify this sentence: we meant both constrained biases for 1 and 2 inputs.

**RC: L310: "higher likelihood of errors being introduced in the input data" $\rightarrow$ What kind of errors are you talking about here? The input data is taken from a model simulation where all data points should be consistent with each other. Except for numerical precision, these data should not have a considerate uncertainty as for instance satellite observations. Please clarify.**

AR: We will clarify this sentence: we meant while cycling the surrogate model, we introduce more errors at each iteration.

**RC: L310-316: "As we cycle the neural network, ..." $\rightarrow$ Do you want to say that the 2-input NN is able to represent a higher degree of nonlinear physics and therefore shows more chaotic behavior?**

AR: We will clarify this sentence, we simply meant that in the inference stage, adding more errors in the neural network logically lead to an increase of the errors in the output of the neural network.

**RC: L 324-326: "The consistent performance of this model across different evaluation metrics and scenarios further validates its reliability and 325 robustness. This surrogate configuration is able to capture the essential features and patterns of SIT dynamics, enabling more accurate predictions compared to other configurations." $\rightarrow$ Please add references to the Tables and Figures with the results you are referring to.**

AR: Thank you for your remark. We will add the references in the sentence.

**RC:** Figure 4. → Please add which NN are displayed with or without constraint.

**AR:** Thank you for your remark. We will add in the caption to specify this is the constraint surrogates.

**RC:** L334: "in" → on

**AR:** Thank you for your remark. We will correct the sentence accordingly.

**RC:** Table 4: 1) Please add at which lead time these statistics are computed. 2) Fig. 5 → Fig. 5a

**AR:** Thank you for your remark. We will add the lead time, those RMSEs are computed after 15 days.

**RC:** Figure6. "surrogate model" → which of the four models is actually shown here?

**AR:** We will add in the caption, the surrogate model shown is the constrained NN with 1 input, note that in L315, we mention that in the rest of the manuscript, we will focus on the 1 input NN.

**RC:** Figure 6. "The trajectories for 30 days are shown in red for neXtSIM and yellow for the surrogate model." → Please use different colors to not confuse them with the ice extent plotted in the same colors in subfigure a) and b).

**AR:** Thank you for your remark. We will change the color scheme accordingly.

**RC:** L.354-360: "In order to verify this visual impression,..." → This entire paragraph requires a more in-depth analysis. What is the separation of the two trajectories over time, etc. Also, more than just the four trajectories would be helpful to better quantify these errors.

**AR:** Thank you for your remark. We believe this 'small' analysis really makes sense to show the advection capability of the surrogate model, we can provide a more 'in-depth' analysis in the appendix, and we will include more trajectories.

**RC:** L 356. "important crack" → What is important about the crack?

AR: It was meant as 'big and persistent', which makes it easy to track over one month

RC: **L 360: "but these differences do not indicate incoherent or erratic behavior."** → **Unclear what is meant by this. The deviations are errors?**

AR: The deviations are indeed errors, but the trajectories are still following coherent paths. We will correct the sentence.

RC: **L363-365: "The observation of a smoothing effect on fine-scale features which increases with the forecast lead time aligns with our expectations"** → **This sentence is unfortunately formulated in a misleading way. If properly forecasted fine scale feature would improve the forecast skill. Only if assumed that the model is unable to properly place the features a smoothed forecast might outperform the fine-scale forecast.**

AR: Thank you for your remark. The sentence was indeed misleading; we will correct it.

RC: **L371: "8"** → **Fig. 8**

AR: Thank you for your remark; we will correct this typo.

RC: **L374: "important"** → **important for what?**

AR: AR: We will correct the sentence. We wished to say that the smoothness increases, as the number of iterations of the surrogate increases.

RC: **L.375: "decrease"** → **decreases**

AR: Thank you for your remark, we will correct this sentence.

RC: **L379-380: "We hypothesize that the neural network has attained its resolution capacity for a correct advection of the sea-ice on the global scale by reducing the fine-scale dynamics that is inherently chaotic and stochastic."** → **It is unclear to me what is meant by this sentence and how it would lead to more structure in the forecasted ice fields. Please elaborate.**

AR: Thank you for your remark. We will add more details in the revised manuscript. We meant that the surrogate model is able to correctly advect sea-ice thickness until a given resolution, over which by smoothing the fine-scale dynamics give better forecast fields in terms of RMSE.

RC: L 383: "arctic" → Arctic

AR: Thank you for your remark; we will correct this sentence.

RC: L386: "initialization periods evenly distributed during that period" → Be more specific: initialized every month?

AR: Thank you for your remark; we will be more specific: forecasts are initialized every 30 days.

RC: L. 387: "In the appendix see Fig. D3" → Does not fit to the rest of the sentence, please rewrite.

AR: Thank you for your remark; we will correct the sentence.

RC: L 387: "propose" → ?

AR: We will correct the sentence; we meant 'show'.

RC: L.389: "In the bottom panel of the figure" → Which figure are you talking about? Fig.D3 does not show global average SIT. . .

AR: The mention of the snapshot in the sentence above is misleading, we will correct the sentence, we meant Fig. 9.

RC: L391-401: "This consistency. . . " → This paragraph is hard to understand and the described hypothesis is hard to follow. Please clarify and add a more comprehensive analysis to justify your points raised.

AR: We will add a more comprehensive explanation, but the main idea is that by having a really low bias, at 12 hours, the surrogate model is correctly able to follow on the long-term the global dynamics of the SIT.

RC: L400-401: "As anticipated, the surrogate model performs significantly better than persistence during periods of high variation, particularly during summer and autumn." → What is in

the other seasons? From Fig.10 it looks like the surrogate model only clearly outperforms persistence from August/September to January, while in spring there seems to be no skill. Please elaborate on this and clarify in which periods there is no gain over persistence.

AR: As mentioned in the major comment, in Fig.10, the variable exhibited is the SIE, which is post-processed from SIT, and in which we only get the accuracy of the position of the MIE. In other seasons where the ice edge is not moving much, the persistence gives a good baseline for the position of the ice edge, and while our surrogate model still correctly advects SIT, the results for the SIE accuracy are not really different from persistence. We will clarify this point in the revised manuscript.

RC: L406: "This opens the perspective to run a large ensemble of simulations for complex sea-ice models, which can facilitate data assimilation." → Please discuss how this fits to the smoothening effect of the model. It might be hard to create an ensemble spread if the model blurs all features. (See major comment above)

AR: This is a relevant point. Our main idea in this sentence was the computational cost. The smoothening might cause a collapse, but the development of better surrogate models might solve this issue.

RC: Figure 8 caption: "blue" → orange?

AR: Thank you for your remark; we will correct the sentence.

RC: L411: "has reached its resolution capacity for correctly simulating the advection of sea ice on a global scale." → This sentence appears the second time in the text and it is unclear to me what is meant with it.

AR: We will modify the sentence, we meant that the surrogate model is good at advecting sea-ice thickness until an efficient resolution under which it diffuses the field.

RC: L417: "This hypothesis implies that the surrogate model focuses on capturing the dominant advection patterns that drive the overall behavior of sea ice, while sacrificing some of the finer details." → This sounds a bit too active for a computer model to me. Isn't that focus determined by the researchers defining the

**cost function that the model aims to minimize while training? Please comment on strategies how to overcome this issue, e.g. new loss functions, more training data, etc. Or do you think a NN is unable to reproduce these fine-scale features at all? (See major comment above)**

AR: Thank you for your remark. As mentioned above, deterministic neural networks won't be able to significantly improve this behavior. Stochastic neural networks have better chances to overcome this issue. We will clarify this in the new discussion part.

RC: **L 425: "have important information for the prediction from the physical model" → Unclear what is meant with this!**

AR: This sentence was indeed unclear. We will correct it, we meant that those variables are important for sea-ice models itself, as they are prognostic variables, and might offer some insights about the physical model.

RC: **L432: "instantiation" → ?**

AR: We will correct this sentence. We meant that the results are different, especially since neXtSIM simulation was forced with ERA5, and that we are trying to predict SIT with fundamentally different forcings.

RC: **L436: "similarly simulated" → The simulated fields are very smooth and hardly similar in nature to the feature-rich fields that neXtSIM is capable of simulating.**

AR: Indeed this sentence is misleading; we will change it.

RC: **Appendix C1 "Partial Convolution algorithm" → Here seems to be text missing.**

AR: We will add a more in-depth description of the algorithm.

**References**

Finn, T. S., Disson, L., Farchi, A., Bocquet, M., and Durand, C. (2023). Representation learning with unconditional denoising diffusion models for dynamical systems. *Nonlinear Processes in Geophysics*.

---

## Author Response (AR1)

**Author's Response**
**for 'Data-driven surrogate modeling of high-resolution sea-ice thickness in the Arctic'**

Charlotte Durand, Tobias Sebastian Finn, Alban Farchi,
Marc Bocquet, Guillaume Boutin and Einar Òlason

January 2024

**RC: Reviewer Comment**; AR: Author Response

**1  Response to Referee 1**

**RC:** In this manuscript, the authors present a machine-learning based surrogate model of the numerical sea-ice model neXtSIM. The presented surrogate model simulates sea-ice thickness and its predictions outperform the climatology benchmark for lead times up to 8 months. The findings of this study are a valuable addition to the field and illustrate how machine learning can be used to reduce the computational costs of sea-ice simulations, e.g. in the context of ensemble forecasting. The main shortcoming of the surrogate model presented, however, is that it simulates very smooth thickness fields compared to the feature-rich neXtSIM input data used for training that includes for example leads. The authors address and analyze this issue in the manuscript, but I still have the major comments outlined below regarding the presentation, analysis, and interpretation of this point that should be addressed before I can recommend this paper for publication.

AR: We deeply appreciate the reviewer's thorough and insightful review of our work. In the following, we discuss the raised concerns and what we have changed in our revised manuscript.

**RC: Major comments:**
**Smoothness of simulated ice thickness fields:**

RC: The authors use simulations of neXtSIM, known for its ability to resolve deformation features and heterogeneous sea ice fields, to train the NN emulator. The surrogate model presented in this manuscript is not able to retain these features over multiple iterations and quickly smoothes the sea ice fields, resulting in thickness fields much more similar to coarse-resolution sea-ice simulations. The authors argue that the smoothed version better minimizes the RMSE (MSE used for training) compared to a model that retains these features and potentially gets penalized for misplacing them. Therefore, the model learns to predict smooth fields and mimics large-scale circulation. While this makes sense in light of the cost function used, I have the following issues with how this fact is presented and interpreted:

RC: Why do you use high resolution in the first place? In the abstract and introduction, the authors make the valid point that current models that simulate small-scale features, like e.g. leads, are computationally very expensive and that a surrogate model would be of great benefit here. However, the presented surrogate model is not able to simulate these fine-scale features, but "just" the large-scale dynamics. Now, I am wondering why it is necessary to use the feature-rich simulations for training. In the introduction the authors suggest that "small-scale effects have an advantage on representing the thermodynamics of sea ice", but I am not aware of modeling studies that have proven this point comprehensively. Now I am wondering if the same results could also be achieved with a coarse resolution model that also resolves the large-scale circulation (and is much cheaper to run). Once having trained the NN on coarse resolution model output, the authors should comment on whether there is an additional benefit in using the high-resolution input data that is currently used.

AR: Our justification for training a surrogate model in high resolution was added to the discussion section, as well as in the appendix, in section D1, with the insertion of the Fig. 1. Training a surrogate model for a coarser resolution is faster, however, a surrogate model for high resolutions can resolve more processes. To showcase this in the following, we displayed results for additional experiments on a coarse-grained dataset with $128 \times 128$ grid cells compared to $512 \times 512$ grid cells at the high resolution. The neural network has the same configuration as the high resolution dataset, and follows the same training procedure. To compare the surrogate trained on the high-resolution data to the surrogate trained on the aforementioned coarse resolution, we present the root-mean-squared error (RMSE) depending on the resolution in Fig. 1. The RMSE of the high-resolution

surrogate model is decreased by 31% after 12 hours compared to the coarse surrogate. This improvement similarly holds throughout resolution levels and also for longer lead times. Since this improvement results out of different modelled resolutions, it can be linked to a better representation of the small-scale dynamics and their impacts on larger scales for the high-resolution surrogate. While the training and inference of this high-resolution surrogate model are significantly more time-consuming than the coarse-grained surrogate, it still maintains a rapid processing speed compared to physical model simulations. Consequently, we see a gain by using high-resolution data for surrogate modelling, even if the surrogate leads to smoothed out forecasts.

RC: **Are the smoothed fields really simulating large-scale dynamics and are the presented methods sufficient to show this? The authors compare their NN results against persistence and find increased skill of the surrogate model. They attribute this skill to the fact that the model learned the large-scale dynamics. In Fig. 10 one can see that the model only outperforms persistence forecasts in periods of rapidly changing ice cover and thickness (melt and early freeze period). Couldn't we find the same behavior as well, if the model learned the climatology of ice thickness and relaxes the input sea ice thickness to this climatology? This would be a better benchmark to beat to justify the authors' claim of learned physics. In general, there should be a more in-depth analysis to demonstrate that the model learned large-scale dynamics, for example, how does the integrated ice edge error (Goessling et al., 2016) varies for different lead times (a quantitative analysis of the qualitative comparison in Appendix D), or a more quantitative evaluation of ice drift started in Fig. 6. Another possibility would be to compare the model skill at different spatial scales by e.g. coarse-graining the predictions. Currently, metrics based on pixel values are shown that always include the effect of missing features. In a scale-dependent analysis, the authors could see up to which scale the model has improved skill and if the large-scale variations are represented appropriately. Or can similar information be extracted from your power spectrum analysis?**

AR: A specific paragraph with the better analysis of the SIE results is proposed in Sec. 5.4. Even though it is a useful marker for the marginal ice zone (MIZ), this post-processed variable is inadequate to represent large scale dynamics of SIT, e.g., in the Central Arctic. It only compares the position of the ice edge, and removes the information about global motion of sea-ice thickness inside the ice sheet. A better definition of the $\mathrm{acc_{SIE}}$ is provided in Sec. 4.2, with an explanation to its link to the Ice Integrated Edge

[Figure]

Figure 1: Evaluation of surrogate model RMSE at various resolutions. Two surrogate models trained on high resolution (solid line) and coarse-grained resolution (dashed line) are assessed in terms of RMSE, in comparison to neXtSIM, at different scales and lead times: after 1 iteration (round markers), after 10 iterations (star markers), and after 20 iterations (hexagonal markers).

Error. the variable under consideration for the evaluation of long-term forecast is an accuracy based on sea-ice extent (SIE). We define the SIE with a SIT threshold of 10 cm and then define an accuracy. As written in the paper : *By obtaining a classification mask between ice and no ice, we can easily define an accuracy metric based on the SIE. We define two terms: the first one $N_{>\sigma_{\mathrm{acc}}}$ indicates the number of pixels where $\mathbf{x}_{t_n+k\Delta t}$ and $\mathbf{x}^{\mathrm{f}}_{t_n+k\Delta t}$ disagree on the presence of sea ice, and the second one $N_{<\sigma_{\mathrm{acc}}}$ where the models disagree on the presence of open water. The accuracy is averaged over all $N_s$ samples:*

$$\mathrm{acc_{SIE}}(k) = \frac{1}{N_s} \sum_{n=1}^{N_s} \left( 1 - \frac{N_{>\sigma_{\mathrm{acc}}}(t_n + k\Delta t) + N_{<\sigma_{\mathrm{acc}}}(t_n + k\Delta t)}{N_{\mathrm{sea-ice\,pixels}}} \right). \quad (1)$$

This definition is actually extremely close to the Integrated Ice Edge Error (IIEE), the main difference being that our threshold is based on SIT and not on the sea-ice concentration (SIC). Secondly, regarding the justification of the learning of the physics, we added a paragraph in the discussion. Our surrogate can conserve the average sea-ice thickness over a full year, while it also can represent advection. This indicates that our surrogate model can capture the large-scale dynamical and thermodynamical evolution of the sea ice over the full year. These phenomenons are driven by external forcings from the atmosphere and ocean. Yet, the surrogate model can represent the influence of the forcing on SIT, something that a climatology and, especially, a persistence forecast cannot exhibit. Furthermore, for short-term forecasts, the surrogate model consistently outperforms persistence and the daily climatology and it shows better forecasts than a daily climatology for more than 6 months in terms of forecast skill.

**RC: If smooth fields are better for prediction, why should the scientific community then at all pursue developing feature-rich models like neXtSIM in a prediction context?**

AR: This is just a first step in surrogate modeling of a rich model like neXtSIM, and we can hope that more advancement would bring more fine-scale features even at longer lead times. Smooth fields are good in terms of RMSE. Caused by the discrete-continuous sea-ice processes, this might not be an optimal evaluation metric. Training for other metrics can become increasingly more complicated and would exceed the scope of the study. Furthermore, the surrogate model is statistical driven, whereas models like neXtSIM are based on our physical understanding. Such models based on physical principles can have an advantage especially for futures cases where we have an extrapolation task, caused by climate change. We discussed this point in Sect. 6.

[Figure]

Figure 2: Time evolution of the IIEE over 2018 for the surrogate model at different lead time and daily climatology. IIEE scores are based with neXtSIM as the ground truth.

**RC:** **The authors suggest that the surrogate model will be of great advantage in computing the adjoint model in variational data assimilation or generating larger ensemble sizes. I have four comments on this: 1) The surrogate model smoothened the input fields and by doing so will reduce the spread of an ensemble, potentially limiting its use for data assimilation. 2) Given the smoothness of the simulated thickness fields and the strong differences with the feature-rich input fields, do you think the surrogate and the numerical model are similar enough to use the surrogate for the adjoint, especially over longer assimilation windows? 3) Does using the surrogate model as adjoint work that easily given the interpolation from unstructured to regular grid? 4) To properly use the surrogate model in data assimilation for both creating an ensemble or the adjoint, more model variables should be simulated than just sea ice thickness. Please comment on all these points in the manuscript.**

AR: Thank you for your feedback. We appreciate your remarks, and we incorporated more details on these points in the discussion section. Regarding your first point, it is a valid consideration, and a sentence was added in the discussion. Yet, it is important to note that our initial emphasis will be on variational data assimilation, we do not necessarily think of employing ensembles. Note however that most ambitious implemented 4D-Var are in practice based on approximate adjoints, or exact adjoints of a simplified model.

On the second point, our current approach involves using the surrogate model primarily for short-term forecasting. Despite the smoothing, we anticipate obtaining meaningful results and believe that the utilization of the adjoint could still prove beneficial. This remark was added to the discussion.

Concerning the third point, this is a relevant point. The definition of an adjoint is meaningful concerning the sea-ice thickness field on the projected grid. While performing variational data assimilation on this grid poses no issues, it is important to acknowledge that it might involve a different adjoint compared to the original triangular mesh. However, a common approach in operational variational data assimilation is to use inner and outer loops. In inner loops, cheaper surrogate models, e.g. the model at a lower resolution, are used, whereas the full high-resolution model is only run in the outer loops. Compared to the raw minimization of the variational cost function by employing the full model, there is no guarantee of convergence. This methodology is nevertheless used with much success and might also also a way for such hybrid approaches. This remark was added to the discussion.

On the fourth point, we completely agree. Our upcoming objectives include the training of a multivariate surrogate model, which can be also used for data assimilation. This remark was added to the discussion.

**RC:** **Is it in general not possible to achieve a higher degree of details in the ice thickness or is it your outlined training and network architecture that hinders it? In the introduction, you state that the cost function plays a major role here, but the manuscript lacks suggestions how to potentially overcome this issue. Please outline potential ways forward in the paper.**

AR: The smoothing effects are highly related to the training of the neural network as deterministic surrogate model. We can expect less smoothing for better models as the uncertainty is decreased. However, caused by the availability of the training data and by computational considerations, we focus on predicting lead times of 12 hours, and there are always futures that cannot be predicted with the available data. By maintaining a deterministic neural network, there will consequently be always a smoothing effect, and we generally do not anticipate significant improvements in the quality of details. By adapting generative neural networks, however, the surrogate model could learn to properly sample from the possible futures and achieve a higher level of fine-scale dynamics. It is worth noting that generative models are currently under investigation by different teams for diverse geoscientific problems Finn et al. (2023). Nonetheless, the training of such models is notoriously more difficult than for deterministic models. We expanded on this in the discussion part of the manuscript.

RC: **Text quality: The manuscript follows a clear structure, but the text is in passages hard to read and follow and clearly requires further editing. In parts, words are missing or sentences are half-finished. In times of automated language editing tools, more thorough language editing is possible also for non-native speakers, and I highly encourage the authors to make use of these tools in the future.**

RC: **L27: "of the Arctic" → Consider removing "of the Arctic" as neither CICE nor SI3 are limited to the Arctic**

AR: We removed those terms from the sentence.

RC: **L28: "road" → route?**

AR: We corrected this term, it comes from a bad translation from french.

RC: **L32: "Divergent features in the ice, like leads and polynyas" → Polynyas are not necessarily formed by divergence.**

AR: Thank you for your remark, we modified the sentence accordingly, by removing 'polynyas'.

RC: **L34-35: "Consequently, models correctly representing the effects of such small-scale can have also an advantage in representing the thermodynamics of sea ice." → Could you please add references, on which studies you base this general statement? In my eyes, it is still an ongoing research question if and what advantage these directly resolved small-scale features have in contrast to parameterizations currently used in climate models.**

AR: We agree this is an ongoing research problem and there is no citation available to make this statement, we modified the sentence in this sense to make it clearer.

RC: **L34: "small-scale" → features? Processes? A word seems missing here.**

AR: Thank you for your remark, indeed, we meant 'processes' and we corrected the sentence accordingly.

RC: **L39: "benefit" → benefits**

AR: Thank you for your remark, we corrected the sentence.

RC: **L60: "Explained differently, the surrogate model is trained to reduce errors" → I do not see how this explains the sentences before differently, it basically says the same as the first sentence in L59. Please clarify.**

AR: Thank you for your remark, we removed this sentence as the first sentence is already clear enough.

RC: **L65: "learn" → train?**

AR: Thank you for your remark, we meant 'train' and we corrected the sentence accordingly.

RC: **L72: "surrogate model" → simulate?**

AR: Thank you for your remark, we meant 'emulate' and we corrected the sentence accordingly.

RC: **L84: "model area" → It is not clear if this is the area of the neXtSIM simulations or of the NN. Please clarify the text accordingly.**

AR: We clarified the sentence, we meant the simulations area.

RC: **L106: "Because these forcings are also to guide the neXtSIM simulations" → From this statement I assume that the neXtSIM simulations are also forced with ERA5. Please clarify this already earlier on in the text to prevent confusion.**

AR: We clarified this in section 2.2, neXtSIM simulations are indeed forced with ERA5.

RC: **L131: "add to the inputs the SIT" → Didn't you write above that SIT is already an input? Why add it again? Please clarify.**

AR: This sentence meant that we also included in the training dataset SIT and forcings at time $t - \Delta t$, we made this addition more explicit in the sentence.

**RC: L133-134: "there are called later 'with 2 inputs'. Otherwise, the neural networks are trained 'with 1 input' "** → **Could you please add those labels for clarity to Table1.**

AR: Thank you for your remark, we added those labels to the Table for clarity

**RC: L152: "(Rampal et al., 2019)"** → **This reference is somehow misleading at it is not clear that it only refers to the multi scale features in sea-ice dynamics, and not to the ability of CNNs to represent those. Please clarify this, or remove the citation here.**

AR: Thank you for your remark, this citation is indeed misleading, and we corrected the sentence accordingly.

**RC: Figure 2 Caption: "512, 256, and 128,"** → **The figure also shows images of size 64. Please correct.**

AR: Thank you for your remark, there is one additional layer of depth in the figure, we corrected it accordingly.

**RC: L165: "sea"** → **ocean?**

AR: Thank you for your remark, we meant "ocean" and we corrected the sentence accordingly.

**RC: L188: "global mean of x and y"** → **As x and y have a physical meaning, it would be helpful for readers if you could also write what the local and global loss mean with respect to sea ice, e.g. local and global trends in sea ice thickness.**

AR: Thank you for your remark, we added this clarification in the sentence.

**RC: L192: "$\lambda$ is manually tuned to 100"** → **What do you optimize for, how do you manually decide on best performance? Please clarify.**

AR: Thank you for your remark, we added the detail of the tuning choices in the manuscript, in the appendix, section C3.2. The choice was made with an evaluation on the validation dataset, for several values of $\lambda$, of the forecast skill and the bias. Non-chosen values for $\lambda$ were not evaluated on the testing dataset. The change in the value of $\lambda$ did not have a significant influence on the forecast skill, except for $\lambda = 1000$, in which case there is

an important decrease of the forecast skill. The influence of $\lambda$ was more pronounced on the bias error and has guided our choice towards $\lambda = 100$.

**RC: L224-225: "over all pixels (i, j) of the field of size (Nx , Ny )"** → **Also over land pixels? Including land pixels in the RMSE will artificially reduce its value.**

AR: Yes, we compute the RMSE over all pixels, but since it is also done for every baseline, this artificial reduction does not change the meaning of the results. Yet, we corrected this, and modify accordingly all the figures and tables.

**RC: L235-237: "We define two terms: the first one $N > \sigma$acc indicates the number of pixels where xtn+k$\Delta$t and xftn+k$\Delta$t disagree on the presence of sea ice, and the second one $N < \sigma$acc where the models disagree on the presence of open water." → This is not clear to me. For all pixels, where the two masks disagree, one will show ice and the other will show open water. Shouldn't therefore not also both terms be the same? Please check those definitions and clarify.**

AR: As mentioned above, this is actually really similar to the IIEE definition, with the overestimation term and the underestimation term. We clarified this in the paragraph as it is indeed misleading.

**RC: L250: "kx and ky"** → **Please rename the indexes x and y to not confuse them with the input of the model x and the output y.**

AR: Thank you for your remark, we corrected the indices to avoid the confusion, by using $k_h, k_v$.

**RC: L254: "justified"** → **caused?**

AR: Thank you for your remark, we changed the word 'justifed' for better clarity.

**RC: L259-260: "In practice, this exponent can be numerically estimated by a linear regression between lnE and $ln|k|$." → multiple studies show that linear fits in double logarithmic plots are not ideal for determining power-law exponents, e.g. Clauset et al. (2009). Please elaborate why you chose this method. Also how does you metric takes into account if such a scaling actually ex-**

**ists or not, or are you computing exponents regardless of the distribution?**

AR: Thank you for your remark; we were not aware of those studies, and chose this method for its simplicity. We corrected this computation with a better alternative. We are also computing the regression score for this power law, which were checked, and are systematically high, for the short-term forecast. We changed to the method mentionned in Clauset et al. (2009), and added a description of this computation in the appendix, in section D5. The value of the coefficient does not change much in absolute value, but, as the lead time increases, with a flattening of the curve on the small scale, the spectral exponent decreases less than with the previous computation, which has more sense with our analysis.

RC: **L 270: "in" → on**

AR: Thank you for your remark; we corrected the sentence accordingly.

RC: **L281-282: "However, the impact of the global Eq. 6) on the RMSE relatively small compared to the influence of including additional time steps." → This sentence does not fit to the observed results. When adding the constraint to the 1 input NN the global RMSE reduces by an order of magnitude (as written in the sentence before). Comparing both unconstrained NNs, the global RMSE reduces to about 25% when including additional time steps, so a much lower reduction compared to including the constraint.**

AR: Thank you for your remark. This analysis was related to the RMSE and not the global RMSE, for which it remains true. We made this more evident in the sentence.

RC: **L. 283: "The impact of adding.." → Just "Adding..."**

AR: Thank you for your remark; we corrected the sentence accordingly.

RC: **L288: "reduce" → the RMSE increases!**

AR: Indeed we meant increase... we correct this in the manuscript.

RC: **L 289: "surrogate RMSE" → what is the surrogate RMSE?**

AR: It meant the RMSE of the surrogate, but this 'surrogate' work is indeed misleading, we removed it for better clarity.

RC: **L291-292: "after 12 hours, the global RMSE has improved by a factor 9.4 for the one input surrogate" → Please comment why there is so little improvement for the 2 input NN.**

AR: The global RMSE, with a smaller value than the RMSE, during the training, is much more volatile. While the training takes into account the global loss, and try to reduce it, in the case of the 2 inputs NN, the best model, according to validation loss (the sum of the local and global losses, weighted by $\lambda$), add a given global RMSE which happened to be similar to the model trained without the constant. The fact that the results are still better in constrained case after 15 days, is also a proof that we indeed learned from the global loss term. We added these discussion points to the manuscript.

RC: **P 307: "leaded" → ??**

AR: We corrected this sentence; we meant 'could have lead'.

RC: **L308: "both biases" → which biases?**

AR: We clarified this sentence: we meant both constrained biases for 1 and 2 inputs.

RC: **L310: "higher likelihood of errors being introduced in the input data" → What kind of errors are you talking about here? The input data is taken from a model simulation where all data points should be consistent with each other. Except for numerical precision, these data should not have a considerate uncertainty as for instance satellite observations. Please clarify.**

AR: We clarified this sentence: we meant while cycling the surrogate model, we introduce more errors at each iteration.

RC: **L310-316: "As we cycle the neural network, . . . " → Do you want to say that the 2-input NN is able to represent a higher degree of nonlinear physics and therefore shows more chaotic behavior?**

AR: We clarified this sentence, we simply meant that in the inference stage, adding more errors in the neural network logically lead to an increase of the errors in the output of the neural network.

**RC: L 324-326:** "The consistent performance of this model across different evaluation metrics and scenarios further validates its reliability and robustness. This surrogate configuration is able to capture the essential features and patterns of SIT dynamics, enabling more accurate predictions compared to other configurations." → Please add references to the Tables and Figures with the results you are referring to.

AR: Thank you for your remark. We added the references in the sentence.

**RC: Figure 4.** → Please add which NN are displayed with or without constraint.

AR: Thank you for your remark. We added in the caption to specify this is the constraint surrogates.

**RC: L334:** "in" → on

AR: Thank you for your remark. We corrected the sentence accordingly.

**RC: Table 4:** 1) Please add at which lead time these statistics are computed. 2) Fig. 5 → Fig. 5a

AR: Thank you for your remark. We added the lead time, those RMSEs are computed after 15 days.

**RC: Figure6.** "surrogate model" → which of the four models is actually shown here?

AR: We added in the caption, the surrogate model shown is the constrained NN with 1 input, note that in L315, we mention that in the rest of the manuscript, we will focus on the 1 input NN.

**RC: Figure 6.** "The trajectories for 30 days are shown in red for neXtSIM and yellow for the surrogate model." → Please use different colors to not confuse them with the ice extent plotted in the same colors in subfigure a) and b).

AR: Thank you for your remark. We changed the color scheme accordingly.

**RC: L.354-360:** "In order to verify this visual impression,..." → This entire paragraph requires a more in-depth analysis. What is the

separation of the two trajectories over time, etc. Also, more than just the four trajectories would be helpful to better quantify these errors.

AR: Thank you for your remark. We believe this 'small' analysis really makes sense to show the advection capability of the surrogate model, we added more details on our analysis.

RC: L 356. "important crack" → What is important about the crack?

AR: It was meant as 'big and persistent', which makes it easy to track over one month, we corrected the term with 'persistent'.

RC: L 360: "but these differences do not indicate incoherent or erratic behavior." → Unclear what is meant by this. The deviations are errors?

AR: The deviations are indeed errors, but the trajectories are still following coherent paths. we added more details on our analysis.

RC: L363-365: "The observation of a smoothing effect on fine-scale features which increases with the forecast lead time aligns with our expectations" → This sentence is unfortunately formulated in a misleading way. If properly forecasted fine scale feature would improve the forecast skill. Only if assumed that the model is unable to properly place the features a smoothed forecast might outperform the fine-scale forecast.

AR: Thank you for your remark. The sentence was indeed misleading; we corrected it.

RC: L371: "8" → Fig. 8

AR: Thank you for your remark; we corrected this typo.

RC: L374: "important" → important for what?

AR: AR: We corrected the sentence. We wished to say that the smoothness increases, as the number of iterations of the surrogate increases.

RC: L.375: "decrease" → decreases

AR: Thank you for your remark, we corrected this sentence.

RC: **L379-380: "We hypothesize that the neural network has attained its resolution capacity for a correct advection of the sea-ice on the global scale by reducing the fine-scale dynamics that is inherently chaotic and stochastic." → It is unclear to me what is meant by this sentence and how it would lead to more structure in the forecasted ice fields. Please elaborate.**

AR: Thank you for your remark. We added more details in the revised manuscript. We meant that the surrogate model is able to correctly advect sea-ice thickness until a given resolution, over which by smoothing the fine-scale dynamics give better forecast fields in terms of RMSE.

RC: **L 383: "arctic" → Arctic**

AR: Thank you for your remark; we corrected this sentence.

RC: **L386: "initialization periods evenly distributed during that period" → Be more specific: initialized every month?**

AR: Thank you for your remark; we were more specific: forecasts are initialized every 30 days.

RC: **L. 387: "In the appendix see Fig. D3" → Does not fit to the rest of the sentence, please rewrite.**

AR: Thank you for your remark; we corrected the sentence.

RC: **L 387: "propose" → ?**

AR: We corrected the sentence; we meant 'show'.

RC: **L.389: "In the bottom panel of the figure" → Which figure are you talking about? Fig.D3 does not show global average SIT...**

AR: The mention of the snapshot in the sentence above is misleading, we corrected the sentence, we meant Fig. 9.

RC: **L391-401: "This consistency..." → This paragraph is hard to understand and the described hypothesis is hard to follow. Please**

**clarify and add a more comprehensive analysis to justify your points raised.**

AR: We will add a more comprehensive explanation, but the main idea is that by having a really low bias, at 12 hours, the surrogate model is correctly able to follow on the long-term the global dynamics of the SIT.

RC: **L400-401: "As anticipated, the surrogate model performs significantly better than persistence during periods of high variation, particularly during summer and autumn." → What is in the other seasons? From Fig.10 it looks like the surrogate model only clearly outperforms persistence from August/September to January, while in spring there seems to be no skill. Please elaborate on this and clarify in which periods there is no gain over persistence.**

AR: As mentioned in the major comment, in Fig.10, the variable exhibited is the SIE, which is post-processed from SIT, and in which we only get the accuracy of the position of the MIE. In other seasons where the ice edge is not moving much, the persistence gives a good baseline for the position of the ice edge, and while our surrogate model still correctly advects SIT, the results for the SIE accuracy are not really different from persistence. We clarified this point in the revised manuscript.

RC: **L406: "This opens the perspective to run a large ensemble of simulations for complex sea-ice models, which can facilitate data assimilation." → Please discuss how this fits to the smoothening effect of the model. It might be hard to create an ensemble spread if the model blurs all features. (See major comment above)**

AR: We raised this point in the manuscript. Our main idea in this sentence was the computational cost. The smoothening might cause a collapse, but the development of better surrogate models might solve this issue.

RC: **Figure 8 caption: "blue" → orange?**

AR: Thank you for your remark; we corrected the sentence.

RC: **L411: "has reached its resolution capacity for correctly simulating the advection of sea ice on a global scale." → This sentence appears the second time in the text and it is unclear to me what is meant with it.**

AR: We modified the paragraph, we meant that the surrogate model is good at advecting sea-ice thickness until an efficient resolution under which it diffuses the field.

RC: **L417: "This hypothesis implies that the surrogate model focuses on capturing the dominant advection patterns that drive the overall behavior of sea ice, while sacrificing some of the finer details." → This sounds a bit too active for a computer model to me. Isn't that focus determined by the researchers defining the cost function that the model aims to minimize while training? Please comment on strategies how to overcome this issue, e.g. new loss functions, more training data, etc. Or do you think a NN is unable to reproduce these fine-scale features at all? (See major comment above)**

AR: Thank you for your remark. As mentioned above, deterministic neural networks won't be able to significantly improve this behavior. Stochastic neural networks have better chances to overcome this issue. We clarified this in the new discussion part.

RC: **L 425: "have important information for the prediction from the physical model" → Unclear what is meant with this!**

AR: This sentence was indeed unclear and was corrected. We will correct it, we meant that those variables are important for sea-ice models itself, as they are prognostic variables, and might offer some insights about the physical model.

RC: **L432: "instantiation" → ?**

AR: We corrected this sentence. We meant that the results are different, especially since neXtSIM simulation was forced with ERA5, and that we are trying to predict SIT with fundamentally different forcings.

RC: **L436: "similarly simulated" → The simulated fields are very smooth and hardly similar in nature to the feature-rich fields that neXtSIM is capable of simulating.**

AR: Indeed this sentence is misleading; we changed it.

RC: **Appendix C1 "Partial Convolution algorithm" → Here seems to be text missing.**

AR: We added a more in-depth description of the algorithm.

**2 Response to Referee 2**

RC: **The paper presents a strong case of surrogate modeling by using neural networks to emulate the increase in sea ice thickness, however, the paper lacks clarity at several places in the manuscript and requires minor revisions:**

AR: We deeply appreciate the reviewer's thorough and insightful review of our work. In the following, we discuss the raised concerns and what we have changed in our revised manuscript.

RC: **1. There is little information provided on the choice of atmospheric variables considered as forcings. Please provide more evidence from literature on this.**

AR: We added a paragraph in section 2.2, regarding the justification for the use of atmospheric forcings, as mentionned in the response to the referee. Firstly, let's note that for neXtSIM simulations, the atmospheric forcings consist of the 10 m wind velocities, the 2 m temperature, mixing ratio, mean sea level pressure, total precipitation and the snow fraction. We decided to limit ourselves to the first three. Plueddemann et al. (1998) and Kwok et al. (2013), for example, have shown that the sea-ice drift is strongly linked to wind velocities. There is a strong correlation between the atmosphere winds and the sea-ice motion, up to 0.8 in Central Arctic (Thorndike and Colony, 1982; Serreze et al., 1989; Zhang et al., 2000). Those forcings are a good proxy for the advection of the sea-ice, which is also necessary for emulating sea-ice thickness dynamics. Observational studies have shown that *interannual variability in sea ice conditions is caused by the variability in the large-scale atmospheric circulation which locally manifests itself as surface air temperature and wind anomalies*, (Deser et al., 2000; Prinsenberg et al., 1997). Experiments were originally conducted with additional forcings, including sea-surface temperature (SST); however, SST was later excluded because the simulation was coupled with the ocean in this version of neXtSIM.

RC: **2. If the neural network is designed for future forecasting, none of the input features should belong to the same timestep as the target. In case of this paper, all the atmospheric variables are of same timestep whereas like SIT, they should also be up till 't' timestep. You can justify through experiments how the current setting performs better than the one suggested.**

AR: We chose to incorporate 'future forcings' based on the understanding that, in sea-ice modeling, the advection of sea ice is strongly influenced by the forecast atmospheric forcings. Experiments were also performed without those future forcings, up until $t$. The impact on forecast skills was non-negligible, as displayed in Fig. 3. Note that the results presented here are evaluated on the validation dataset. In the simulations on which our dataset is based on (Boutin et al., 2023), neXtSIM is uncoupled from an atmospheric model and uses just ERA5 forcings. In such settings, the atmospheric forcing can be given by forecasts, and, thus, known for the future. Consequently, using future forcings during training is nonrestrictive in terms of its potential operational capability. As those experiments were not conducted on a test dataset, but at a previous stage of the analysis, they were not included in the manuscript. Nonetheless, a remark was added in Sec.2.2 about this choice.

[Figure]

Figure 3: Forecast skills of surrogate models depending on the addition of future forcings. Two surrogate models are evaluated on the validation dataset: one with atmospheric forcings, up to $t$ (red curve) and another with the addition of future atmospheric forcings, up to $t + 12$ hours (blue curve). Averages of the RMSE are shown with solid curves, and their associated standard deviations are outlined with transparency.

**RC: 3. There are some minor errors that should be corrected: UNet by definition is not a convolutional architecture but it is an encoder-decoder Neural Network architecture with skip-connections. There are several papers utilizing LSTM-based UNet or ConvLSTM-based UNet. Andersson et al. did not propose IceNet for SIC prediction. Their work targets SIP predictions which is slightly different from SIC.**

AR: We added references for more LSTM-based UNet or ConvLSTM-based UNet papers, corrected the mention of the IceNet paper to SIP predictions, and the definition of UNet.

**RC: 4. There are several other recent papers that utilize CNN, ConvLSTM and LSTM for SIC predictions. There is not enough convincing argument present on just relying on UNet for the surrogate model. Did the authors try a CNN or ConvLSTM based architecture for surrogate modeling?**

AR: Several architectures have been tested on a coarse-grained dataset ($128 \times 128$ grid cells) to reduce computation costs. Both mentioned ResNet and ConvLSTM structures have been investigated, yielding quite similar results in terms of forecast skills on the validation dataset. From our experimentation, the specific structure of the convolutional neural network does not matter much. Our LSTM-based approach, with a lag of 48 hours, led to satisfactory forecast skills; caused by the high computational costs (441 s/epoch) compared to the UNet (108 s/epoch), we focus on the UNet structure when we moved to the high-resolution dataset. Regarding the ResNet architecture, also implemented with partial convolution, the results were also quite similar, despite higher computational costs (172 s/epoch). Forecasts skills results are presented in Fig. 4 for UNet, ResNet and ConvLSTM. As no extensive study for those different architectures have been conducted afterwards, and the results presented here are on a coarser resolution, not necessarily fully hyper-optimized. Yet, a description of those experiments and our justification towards focusing on the UNet are provided in the discussion section.

**RC: 5. How was 100 decided as the optimal value of lambda? Did you experiment with other values of lambda in calculating the global loss?**

AR: A more thorough description of the choice of $\lambda$ has been provided in the appendix, in the section C3.2. This value was chosen after experimenting with several values. The impact of $\lambda$ on the forecast skill is negligible, while having an important impact on the bias. The value of $\lambda = 100$ was chosen

[Figure]

[Figure]

Figure 4: Forecast skills for different neural network architecture on a coarse-grained dataset ($128 \times 128$ grid-cells) on the validation dataset. (a) Comparison between a ConvLSTM architecture (red) and a UNet architecture (blue). (b) Comparison between a ResNet architecture (red) and a UNet architecture (blue). The solid curves represent the averaged normalized root-mean-squared error (nRMSE) and in transparency is represented its associated standard deviation. The persistence baseline is also indicated in purple (a) and black (b).

based on the evaluation on the validation dataset. Others values of $\lambda$ were not evaluated on the test dataset. Please find the experiments on the validation dataset in Fig. 5. Selecting a value for $\lambda$ that is excessively large could result in a loss of information at the fine scale. A value of $\lambda = 100$ seems to keep a good balance between fine-scale dynamics and global sea-ice thickness. The experiments, being conducted on the validation dataset, were not added to the manuscript.

**RC: 6. What is the timestep used in case of longterm forecasting?**

AR: We added at the beginning of section 5.4 that for the long-term forecast, the NN trained with one timestep was used.

**RC: 7. Did the authors consider using custom loss function instead of partial convolution to incorporate land-mask into the modeling?**

AR: As we discussed in the referee's answer, we have not considered using only a custom loss function instead of partial convolution, thus further experiments were not added to the manuscript. The loss function is already custom and takes the mask into account. Should we use normal convolutions instead of partial convolutions, we would zero-pad land pixels. The effect of the land masses would be then similar to effects of zero padding at image boundaries, which can lead to artifacts (Liu et al., 2018). Con-

[Figure]

Figure 5: Impact of the choice of $\lambda$ on the forecast skill and bias error on the validation dataset. On the left panel is presented the forecast skill, as defined on the manuscript. The solid curves represent the averaged normalized root-mean-squared error (nRMSE) and in transparency is represented its associated standard deviation. On the right panel is shown the bias error, as defined on the manuscript. The solid curves represent the averaged bias error and in transparency is represented its associated standard deviation. Results for several values of $\lambda$ are shown for both panel : $\lambda = 0$ (blue), $\lambda = 10$ (red), $\lambda = 100$ (yellow), $\lambda = 1000$ (green). The persistence baseline forecast skill is also indicated in black.

sequently, by taking the mask only for the loss function into account, we would possibly generate artifacts in regions with a lot of land masses. Additionally, without masking operations, during cycling of the neural network for longer lead times than 12 h, errors could rapidly accumulate on land and lead to physically inconsistent results.

RC: Ref: 1. Ebert-Uphoff, Imme, et al. "CIRA Guide to Custom Loss Functions for Neural Networks in Environmental Sciences–Version 1." arXiv preprint arXiv:2106.09757 (2021).
2. Ali, Sahara, and Jianwu Wang. "MT-IceNet-A Spatial and Multi-Temporal Deep Learning Model for Arctic Sea Ice Forecasting." 2022 IEEE/ACM International Conference on Big Data Computing, Applications and Technologies (BDCAT). IEEE, 2022.
3. Kim, Eliot, et al. "Multi-task deep learning based spatiotemporal arctic sea ice forecasting." 2021 IEEE International Conference on Big Data (Big Data). IEEE, 2021.

AR: Thank you for the references.

**References**

Boutin, G., Ólason, E., Rampal, P., Regan, H., Lique, C., Talandier, C., Brodeau, L., and Ricker, R. (2023). Arctic sea ice mass balance in a new coupled ice–ocean model using a brittle rheology framework. *The Cryosphere*, 17(2):617–638.

Clauset, A., Shalizi, C. R., and Newman, M. E. J. (2009). Power-law distributions in empirical data. *SIAM Review*, 51(4):661–703.

Deser, C., Walsh, J. E., and Timlin, M. S. (2000). Arctic sea ice variability in the context of recent atmospheric circulation trends. *Journal of Climate*, 13(3):617–633.

Finn, T. S., Disson, L., Farchi, A., Bocquet, M., and Durand, C. (2023). Representation learning with unconditional denoising diffusion models for dynamical systems. *Nonlinear Processes in Geophysics*.

Kwok, R., Spreen, G., and Pang, S. (2013). Arctic sea ice circulation and drift speed: Decadal trends and ocean currents. *Journal of Geophysical Research: Oceans*, 118(5):2408–2425.

Liu, G., Shih, K. J., Wang, T.-C., Reda, F. A., Sapra, K., Yu, Z., Tao, A., and Catanzaro, B. (2018). Partial convolution based padding. *arXiv preprint arXiv:1811.11718*.

Plueddemann, A. J., Krishfield, R., Takizawa, T., Hatakeyama, K., and Honjo, S. (1998). Upper ocean velocities in the beaufort gyre. *Geophysical Research Letters*, 25(2):183–186.

Prinsenberg, S. J., Peterson, I. K., Narayanan, S., and Umoh, J. U. (1997). Interaction between atmosphere, ice cover, and ocean off labrador and newfoundland from 1962 to 1992. *Canadian Journal of Fisheries and Aquatic Sciences*, 54(S1):30–39.

Serreze, M. C., Barry, R. G., and McLaren, A. S. (1989). Seasonal variations in sea ice motion and effects on sea ice concentration in the canada basin. *Journal of Geophysical Research: Oceans*, 94(C8):10955–10970.

Thorndike, A. S. and Colony, R. (1982). Sea ice motion in response to geostrophic winds. *Journal of Geophysical Research: Oceans*, 87(C8):5845–5852.

Zhang, J., Rothrock, D., and Steele, M. (2000). Recent changes in arctic sea ice: The interplay between ice dynamics and thermodynamics. *Journal of Climate*, 13(17):3099–3114.